# Aggregating in vitro-grown adipocytes to produce macroscale cell-cultured fat tissue with tunable lipid compositions for food applications

John Se Kit Yuen Jr[1], Michael K Saad[1], Ning Xiang[1], Brigid M Barrick[1], Hailey DiCindio[1], Chunmei Li[1], Sabrina W Zhang[1], Miriam Rittenberg[2], Emily T Lew[1], Kevin Lin Zhang[1], Glenn Leung[1], Jaymie A Pietropinto[1], David L Kaplan[1]*

[1]Biomedical Engineering Department, Tissue Engineering Resource Center, Tufts University, Medford, United States; [2]Biological Engineering Department, MIT, Cambridge, United States

**Abstract** We present a method of producing bulk cell-cultured fat tissue for food applications. Mass transport limitations (nutrients, oxygen, waste diffusion) of macroscale 3D tissue culture are circumvented by initially culturing murine or porcine adipocytes in 2D, after which bulk fat tissue is produced by mechanically harvesting and aggregating the lipid-filled adipocytes into 3D constructs using alginate or transglutaminase binders. The 3D fat tissues were visually similar to fat tissue harvested from animals, with matching textures based on uniaxial compression tests. The mechanical properties of cultured fat tissues were based on binder choice and concentration, and changes in the fatty acid compositions of cellular triacylglyceride and phospholipids were observed after lipid supplementation (soybean oil) during in vitro culture. This approach of aggregating individual adipocytes into a bulk 3D tissue provides a scalable and versatile strategy to produce cultured fat tissue for food-related applications, thereby addressing a key obstacle in cultivated meat production.

*For correspondence: David.Kaplan@tufts.edu

Competing interest: The authors declare that no competing interests exist.

## Editor's evaluation

This paper describes an important new method to cultivate fat tissues in vitro for meat production, by cell aggregation after the adipocytes have fully differentiated in a two-dimensional monolayer. The authors present exceptional evidence that this approach is scalable and that the lipid treatment of the cultured adipocytes modifies their fatty acid composition in the triglyceride as well as the phospholipid portions, which is important for engineering the taste component in cellular agriculture. The work will be of broad interest to bioengineers, tissue engineers, biomaterial scientists, and stem cell biologists.

## Introduction

Cultivated meat (also cultured, cell-cultured, in vitro meat) uses tissue engineering to produce meat for food (*Datar and Betti, 2010*; *Post, 2012*; *Post and Hocquette, 2017*). Canonically, this research has focused on recreating the muscle component of meat (*Post, 2012*; *Benjaminson et al., 2002*; *Simsa et al., 2019*; *Ben-Arye et al., 2020*; *Furuhashi et al., 2021*; *Verbruggen et al., 2018*). However, fat is key to the taste and texture of meat (*He et al., 2020*; *Moritz et al., 2015*; *Frank et al., 2016*; *Frank et al., 2017*). For example, peak evaluation scores were achieved with beef samples containing ~36%

crude fat (*Iida et al., 2015*). Hundreds of volatile compounds are released when meat is cooked, with a majority originating from lipids (*Frank et al., 2017*; *Ba et al., 2012*; *Mottram et al., 1982*). Fat is also responsible for the species-specific flavor of meat, making it important for reproducing the flavors of specific animals (*Ramalingam et al., 2019*). In addition to cultivated meat, in vitro-grown fat could also be used to enhance existing plant-based meats, as complex species-specific flavors of animals are difficult to recapitulate de novo (*Fish et al., 2020*; *van Vliet et al., 2021*; *Bohrer, 2019*).

While cultured fat stands to greatly improve the quality of alternative proteins, producing macro-scale tissues remains a major challenge due to mass transport limitations. Even millimeter-scale in vitro tissues are challenging to generate, as oxygen and nutrients are only able to diffuse through several hundred microns of dense tissue (*Coyle et al., 2019*; *Murphy et al., 2017*; *Helmlinger et al., 1997*; *Jain et al., 2005*). Various approaches have been implemented in tissue engineering to overcome diffusion limitations, such as vascularization and the incorporation of perfusion channels (*Kolesky et al., 2016*; *Sarig-Nadir et al., 2009*; *Li et al., 2018*; *Lovett et al., 2007*; *Lai et al., 2009*; *Muller et al., 2019*; *Kang et al., 2009*; *Louis et al., 2021*; *Volz et al., 2019*; *Wittmann et al., 2015*; *Yang et al., 2020*). However, current strategies are often limited by scalability and complexity. A dense array is required when incorporating perfusion channels, as all tissues must remain within several hundred microns of a channel (*Yamamoto et al., 2012*). For adipose tissue, lipid accumulation decreases considerably once cells are ~1 mm away from a perfusion channel (*Li et al., 2018*). Vascularization of macroscale tissues is achieved via co-culture with endothelial cells, which spontaneously organize into capillary-sized vessels (*Andrée et al., 2019*). However, spontaneous vessel formation may not be feasible for centimeter- to meter-scale tissues, as cultivated cells may die during the time required for the vasculature to form (*Unal et al., 2018*). Capillaries may also be too small to deliver a sufficient volume of media when supporting larger tissues (*Tetzlaff and Fischer, 2018*; *Moya et al., 2013*). Thus, many challenges remain to scale fat (and other) tissues to bulk production levels.

In this study, our goal was to develop a relatively simple method of producing bulk cultured fat tissue that circumvents contemporary mass transport limitations. This was achieved by growing murine and porcine adipocytes in thin layers with easy access to the culture media, followed by post-growth aggregation into 3D adipose after sufficient adipocyte maturation (*Figure 1*). Aggregation at the end of cell culture removed the need for nutrient delivery via vascularization or elaborate 3D tissue perfusion, thus reducing costs and improving scalability. This approach is feasible when creating tissue solely for food purposes, as there is no requirement for continued cell survival once the final edible tissue is produced. This paradigm allows for simpler approaches and outcomes in comparison to regenerative medicine goals.

We hypothesized that aggregating individual adipocytes would be sufficient to reproduce the taste, nutrition, and texture profile of fat, as adipose tissue in vivo is a dense aggregation of lipid-filled adipocytes with a sparse extracellular matrix (ECM) (*Mescher, 2021*). Lipid-filled adipocytes were grown in vitro and aggregated into bulk macroscale tissues within suitable matrices, then compared with native adipose tissue from various animals. Cultured fat and native adipose samples were characterized for lipid droplet morphology and mechanical properties to infer textural characteristics. The fatty acid compositions of cultured adipocytes were also analyzed to garner insight into flavor and nutrition.

## Results

### Murine adipogenic precursor cells accumulate lipid and express adipogenic markers during 2D in vitro culture, resembling native adipocytes after 30 days of adipogenesis

To obtain individual fat cells needed to test the concept of adipocyte aggregation into 3D fat tissue, we cultured murine 3T3-L1 cells using 2D culture for good cellular access to nutrition and simple cell harvest. 3T3-L1s were chosen for initial experiments as they are readily obtainable from cell banks, and because effective fat differentiation protocols for the cell line already exist (*Li et al., 2018*; *Scott et al., 2011*). After reaching confluency, 3T3-L1s were induced to differentiate into adipocytes for 2 days, then switched to lipid accumulation media until 15–30 days of culture (*Figure 2A*). At day 15, cultured adipocytes were 81.6% (±5.5%) positive for peroxisome proliferator-activated receptor gamma (PPARγ) (*Figure 2B*, *Supplementary file 1*). 30-day adipocytes contained ~3 times more

lipid per cell than 15-day adipocytes (*Figure 2C and D*). During cell culture, gentle treatment of the cells (e.g., when changing culture media) was key to preventing lift-off of lipid-filled adipocytes (detailed technique outlined in *Video 1* and the Materials and methods). A positive correlation was observed between adipogenic culture duration and lipid droplet size (*Figure 2E*, *Supplementary file 2*). Mean lipid droplet diameters for 15- and 30-day adipocytes were 8.1 µm (±0.04 SEM) and 11.1 µm (±0.1 SEM), respectively. Cultured adipocytes exhibited the appearance of packed lipid droplets observed when imaging native adipocytes from various animals, particularly that of 7-day old mouse (7D mouse) and chicken (*Figure 2F*).

## Aggregated in vitro adipocytes form macroscale cultured fat tissue constructs

After 15–30 days of adipogenesis, cultured adipocytes were harvested with a cell scraper (*Figure 3A and B*). Adipocytes were scraped mechanically due to insufficient detachment when using Accumax (incubation time: >20 min, data not shown). During cell harvest, cultured adipocytes aggregated by the mechanical action of the cell scraper appeared like fat tissue or lipoaspirate (*Figure 3—figure supplement 1*, *Video 2*). Micrographs of a piece of aggregated adipocytes are available in *Figure 3—figure supplement 2*. We found that one culture flask with 175 cm$^2$ surface area generally yielded ~0.8 g of cell-scraped adipocytes (data not shown). After harvest, scraped adipocytes successfully formed macroscale 3D tissues that subsequently stayed as one mass after mixing with binders to hold individual cells together (*Figure 3C and D*). Sodium alginate and microbial transglutaminase (mTG) as 'generally recognized as safe' (GRAS) materials in the United States were selected as

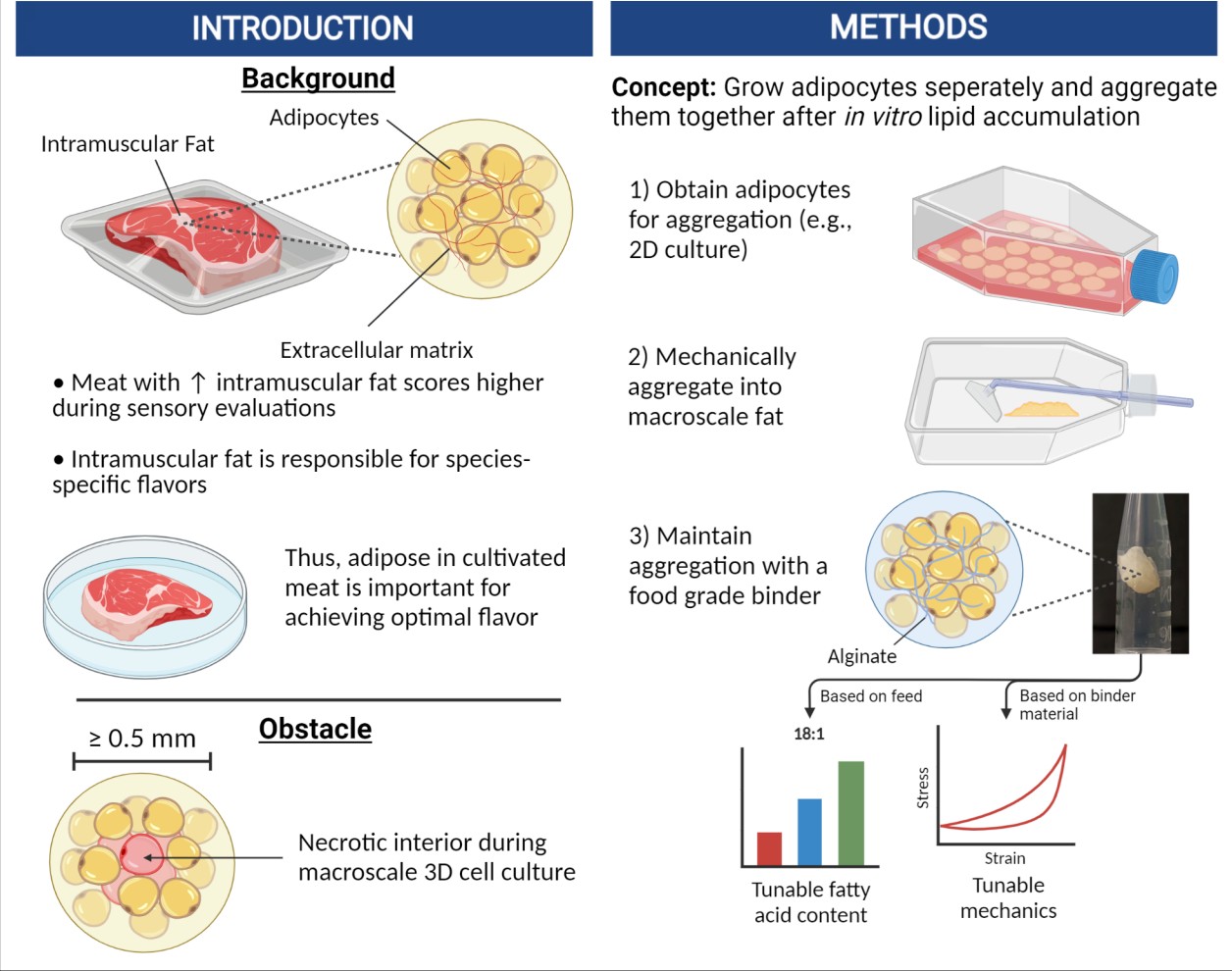

**Figure 1.** Graphical abstract covering the overall concepts behind producing macroscale volumes of cultured fat in this study.

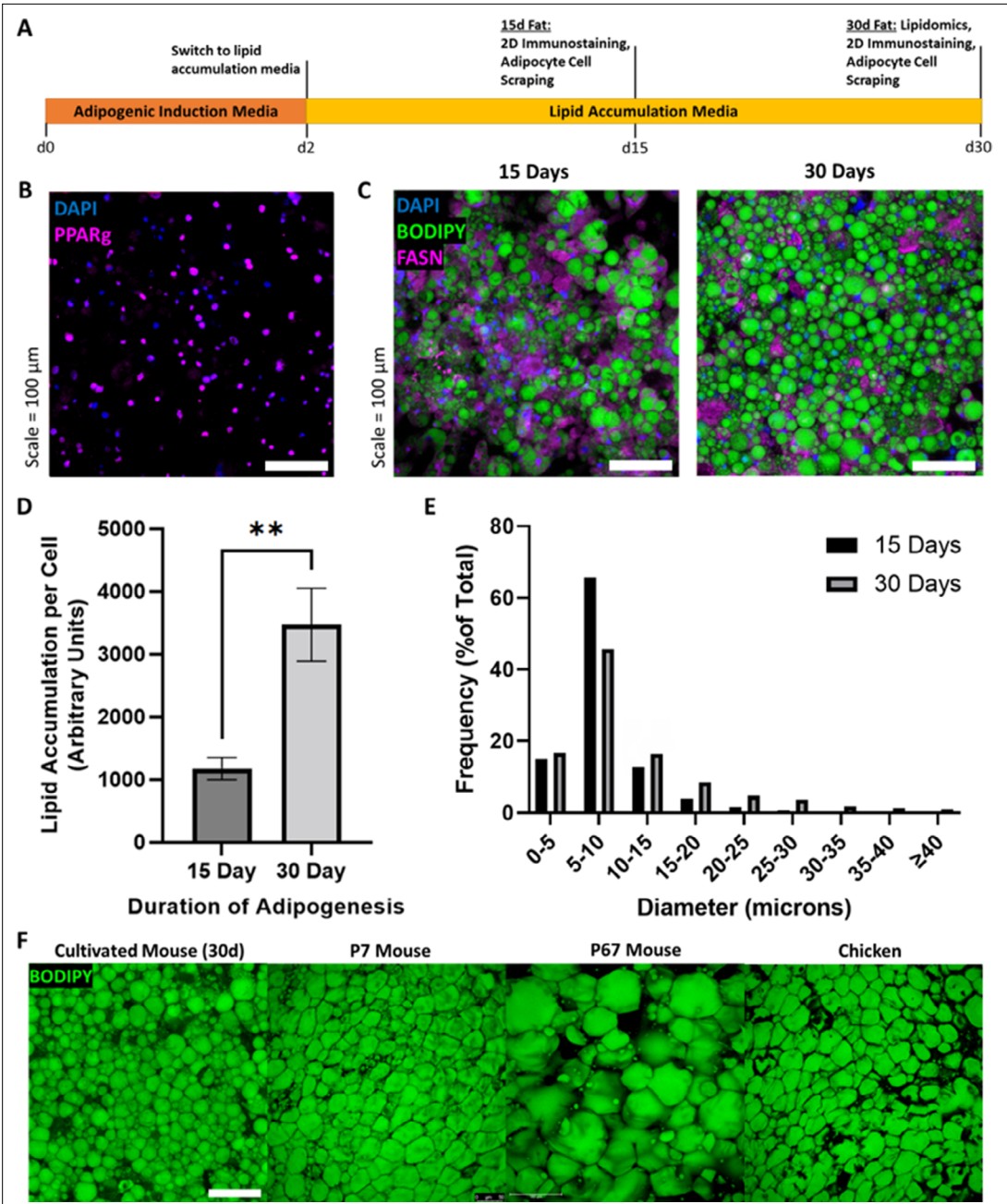

**Figure 2.** Assessment of in vitro murine adipocyte differentiation. (**A**) Timeline of 3T3-L1 adipogenic differentiation. Confluent preadipocytes were grown in adipogenic induction medium for 2 days, then switched to lipid accumulation media for 15 or 30 days, where cells were stained and imaged, or harvested for lipidomics and 3D cultured fat tissue formation. (**B**) 15-day adipocytes stained for the adipogenic transcription factor peroxisome proliferator-activated receptor gamma (PPARγ) (magenta), as well as nuclei via DAPI (blue). (**C**) Lipid-stained (BODIPY, green) adipocytes after 15 and 30 days of adipogenesis. The in vitro adipocytes were also stained for DNA via DAPI (blue) and fatty acid synthetase (magenta). (**D**) The mean degree of lipid accumulation in 15- and 30-day cultured adipocytes, normalized by the number of cells detected via nucleus counting. Sample groups were compared using an unpaired t test with Welch's correction, where p≤0.01 (**). (**E**) Frequency distributions of lipid droplet diameters from cultured adipocytes adipogenically grown for 15 and 30 days, compared via chi-square test in *Supplementary file 2*. (**D**) represents n=4 technical replicates from one experiment, while (**E**) represents n=4 technical replicates from a second experiment. (**F**) Lipid staining images (BODIPY) of 30-day in vitro 3T3-L1s compared to native adipocytes from chicken and mice of two ages. '7D Mouse' and '67D Mouse' refer to 7- and 67-day-old mice, respectively. Scale bar (same for all images) represents 100 μm.

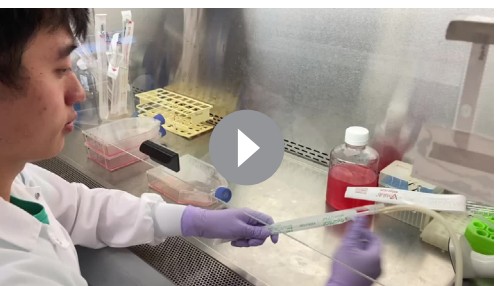

**Video 1.** Methods for minimizing adipocyte cell detachment during cell culture (when changing culture media). Inversion during media (liquid) aspiration (0:06). Dispensing to sidewall of flask during inversion (1:18). Reinverting flask carefully from inverted position (1:42). https://elifesciences.org/articles/82120/figures#video1

binders for cultured adipocytes, with precedent for alginate as a fat replacer and mTG as a binder in meat products (*Figure 3—figure supplement 3*; *ECFR, 1982*; *FDA, 1999*).

## Alginate-based macroscale cultured fat tissues exhibit a similar compressive strength with native fat tissues from cow, pig, and chicken

After forming macroscale constructs, cultured fat tissue samples were subjected to uniaxial compression testing as a proxy for food texture. For alginate-based cultured fat tissues, 1.6% (final concentration) was selected after preliminary experiments (*Figure 3—figure supplement 4*). The degree of adipocyte lipid accumulation (cultured fat tissues produced using 15- versus 30-day-old adipocytes) did not have a significant impact on resistance to compression (*Figure 3Ei, Fi*). Alginate fat tissues containing 15- and 30-day adipocytes exhibited mean compressive stresses of 11.1 kPa (±5.2) and 12.8 kPa (±7.3) at 50% strain respectively (p=0.7304, Welch's t-test). For 15- and 30-day mTG-based cultured fat tissues, the mean resistances to compression were 0.5 kPa (±0.2) and 0.7 kPa (±0.2) at 50% strain, respectively (p=0.1030, Welch's t-test).

The stress-strain profile of alginate-based cultured fat tissues were within the general range of live-stock and poultry fat tissues, while displaying similar hysteresis behavior during loading and unloading (*Figure 3E, ii*). At 50% strain, mean compressive stresses for beef and pork fat tissues were 20.2 kPa (±12.5) and 35.6 kPa (±17.5), respectively, while the mean value for chicken fat tissue was 6.2 kPa (±2.0). mTG-based cultured fat tissues were more similar to rendered fats, although lard and tallow samples appeared stronger with mean compressive stresses of 4.6 kPa (±1.4) and 2.5 kPa (±0.9) at 50% strain, respectively (*Figure 3F, ii*). mTG fat tissues maintained some resistance during unloading, while rendered fats did not rebound after compression.

## The fatty acid compositions of in vitro murine adipocytes can be tuned by lipid supplementation during cell culture

Next, we looked to compare the intracellular lipids of in vitro- and in vivo-grown mouse adipocytes. In vitro adipocytes were cultured with 0–1000 µg/ml of Intralipid (a soybean oil-based lipid emulsion) during the lipid accumulation phase of adipogenic differentiation (d2-d30). Here, the goal was to investigate any positive or negative effects of lipid supplementation on adipocyte fatty acid (FA) composition relative to in vivo adipocytes. Lipid supplementation (e.g., with Intralipid) has been reported to enhance adipogenic differentiation and may thus be a useful strategy in enhancing fat cell lipid accumulation during cultured fat production (*El Hadri et al., 2004*; *Ding et al., 2003*; *Kokta et al., 2008*; *Kim et al., 2015*; *Yanting et al., 2018*; *Malodobra-Mazur et al., 2019*; *Jurek et al., 2020*; *Kim et al., 2020*).

When comparing the intracellular triacylglycerides (TAGs) of in vitro adipocytes (also termed in vitro fats [IVFs]) without lipid supplementation (other than lipids in fetal bovine serum [FBS]) and native mouse fats, the major difference observed was an enrichment of 16:1 and a lack of 18 carbon FAs (*Figure 4A*). Unsupplemented IVFs contained significantly less 18:1 and 18:2 versus 7D and 67D mouse fats, as well as slightly less 18:0 and 18:3 than 67D mouse. 16:0 was observed at similar levels to 67D mouse, while being lower than 7D mouse. Lower carbon FAs were found at higher levels in unsupplemented IVFs (e.g., 6:0), but these FAs were also occasionally enriched in 7D mouse (14:0, 12:0, 10:0). The addition of lipids (500–1000 µg/ml Intralipid) during cell culture altered adipocyte TAGs dose dependently, generally bringing FA compositions closer to that of the native fats largely through a reduction in 16:1 and increase in 18:1. Intralipid-treated IVFs had elevated levels of 16:0 and 18 carbon FAs (18:0, 18:1, 18:2, 18:3), which correlates with the composition of soybean oil,

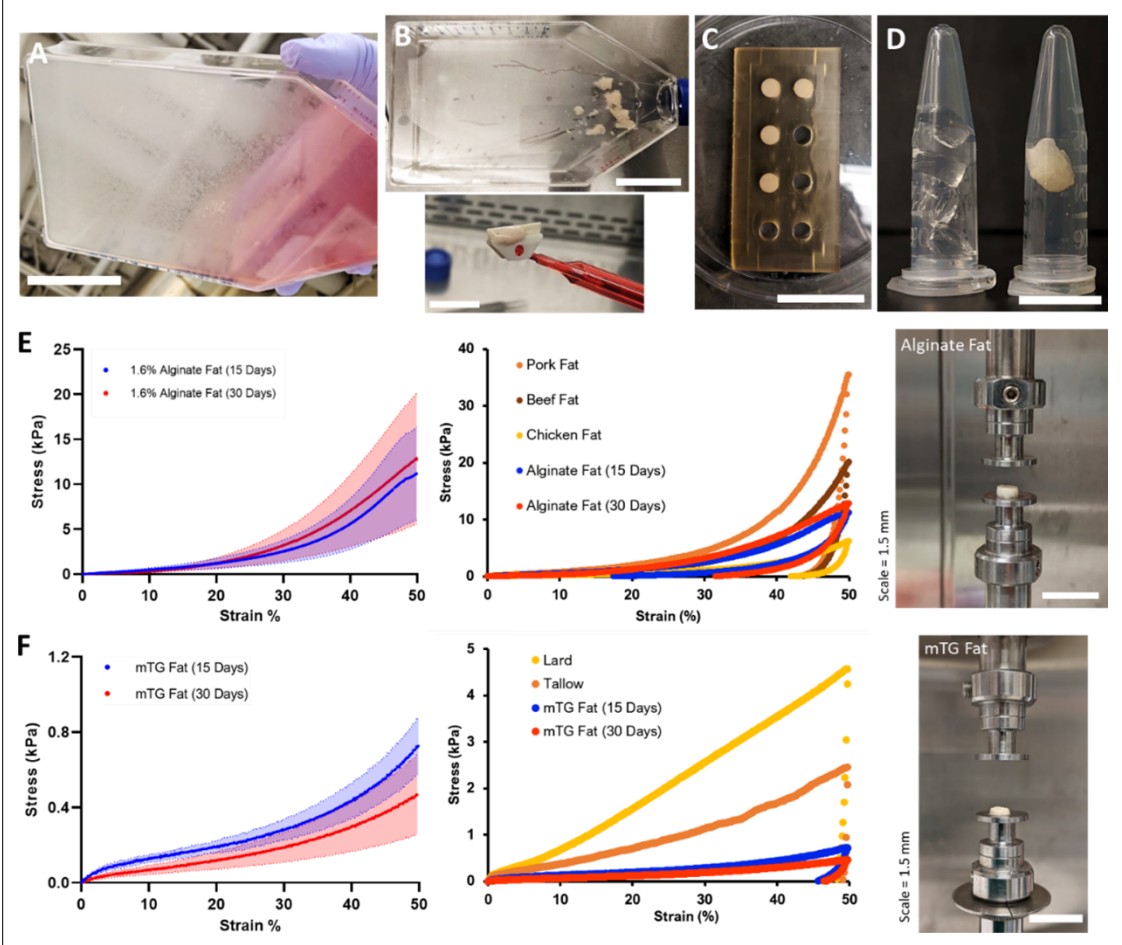

**Figure 3.** Mechanical characterization of 3D cultured fat tissues. (**A–D**) Steps for producing 3D macroscale cultured fat tissues through the aggregation of individual adipocytes grown in vitro. (**A**) *Adipogenesis*: Adipocytes were differentiated in vitro for 15–30 days. Lipid-accumulating fat cells turned the bottom of the cell culture flask opaque. Scale bar 5 cm. (**B**) *Cell harvest*: Adipocytes were mechanically collected with a cell scraper, which aggregated the cells into masses of cultured fat. Scale bars 5 and 1.5 cm for the top and bottom images, respectively. (**C**) *Binding into 3D tissue*: Harvested fat was combined with a binder (e.g., alginate, transglutaminase) in a mold to add structure. Scale bar 3 cm. (**D**) *3D cultured fat*: Cylinders of structured cultured fat tissue after removal from the mold. A piece of 3D cultured fat tissue made with 1.6% alginate is shown in the right tube, while the left tube contains 1.6% alginate without fat cells. Scale bar 1 cm. (**E and F**) Mechanical testing of cultured and native fat tissues (uniaxial compression). (**E, i**) shows the compressive strength of alginate-based cultured fat tissues formed using 15- or 30-day adipocytes, while (**F, i**) shows the same for microbial transglutaminase (mTG)-based cultured fat tissues. Solid lines represent mean values, while the shaded areas represent standard deviations. (**E, ii**) A compressive strength comparison of alginate-based cultured fat tissues with intact pig, cow, and chicken adipose tissues. Data points represent mean values, and the overall data represent tissue loading from 0% to 50% strain over 30 s, followed by unloading to 0% strain over the same duration. (**F, ii**) The same compressive strength comparison as (**E, ii**) but with mTG fat tissues alongside rendered animal fats from pigs (lard) and cows (tallow). (**E, iii**) depicts macroscale alginate-based cultured fat tissues on the mechanical testing apparatus prior to compression. (**F, iii**) depicts the same for mTG-based cultured fat tissues. n=4 for all alginate-based cultured fat constructs. n=3 and 5 for 15- and 30-day mTG cultured fat constructs, respectively. n=5, 6, 7 for beef, pork, and chicken adipose samples respectively. n=4 for lard and tallow samples respectively. All 'n' values refer to technical replicates.

The online version of this article includes the following figure supplement(s) for figure 3:

**Figure supplement 1.** Photographs of aggregated cell-cultured adipocytes after manual harvest.

**Figure supplement 2.** Low magnification micrographs of a group of aggregated adipocytes after manual harvest.

**Figure supplement 3.** Detailed steps for producing macroscale 3D cultured fat constructs from aggregated adipocytes, using slow-gelling alginate or microbial transglutaminase as binders.

**Figure supplement 4.** Uniaxial compression testing of macroscale cultured fat tissues produced from 0.8% and 1.6% (final concentration) alginate containing aggregated lipid-laden in vitro mouse adipocytes (15 days of adipogenesis).

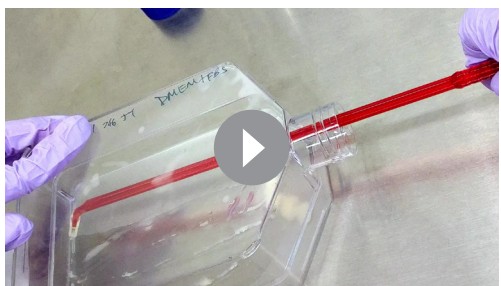

**Video 2.** Harvesting cultured fat by cell scraping in vitro murine adipocytes.

https://elifesciences.org/articles/82120/figures#video2

the major constituent of Intralipid (aside from water) (*Supplementary file 3*; *Abdelghany et al., 2020*). Notably, the increase in 18:3 in lipid supplemented adipocytes may reflect the uptake of alpha-linoleic acid from the Intralipid (*Figure 4E*). Principal component analysis (PCA) shows lipid supplementation bringing in vitro murine TAG compositions closer to their native equivalents, particularly to 7D mouse (*Figure 4—figure supplement 1*). When looking at TAG FA saturation, samples did not vary greatly in terms of monounsaturated FAs (MUFAs), except for 7D mouse fat (*Figure 4C*). Unsupplemented IVFs contained low levels (2.9%±0.1) of polyunsaturated FAs (PUFAs), but this increased to 12.3% (±2.7) with 1000 µg/ml of Intralipid, similar to that of 7D mouse. PUFA increases for Intralipid IVFs were mirrored by a decrease in saturated FAs (SFAs), going from 48.2% (±0.9, unsupplemented IVFs) to 39.9% (±3.0, 1000 µg/ml Intralipid).

In addition to TAGs, we also looked at phospholipid FA compositions due to their potential role in the flavor of meat (*Mottram and Edwards, 1983*; *Huang et al., 2010*). Like the TAGs, unsupplemented IVFs were high in 16:1s (*Figure 4B*). 16:0 levels were high and above that of 67D mouse, but similar to 7D mouse. 18:1 levels were lower than 67D mouse, but higher than 7D mouse. Unsupplemented IVFs contained noticeably fewer 18:0s and 20:4s than native fats. Lipid supplementation (Intralipid) had a muted effect on IVF phospholipids, with smaller changes in 16:1 (–8.3%±1.4) and 18:1 (+3.9% ± 0.6) between 0 and 1000 µg/ml Intralipid-treated cells. No significant changes were observed for 16:0 and 18:2. When looking at FA saturation for the phospholipids, these changes are reflected by a decrease in MUFAs for lipid supplemented adipocytes, accompanied by an increase in SFA content (*Figure 4D*). Lipid supplementation ultimately results in a decrease in overall SFA levels though, as TAGs make up the majority of lipids within adipocytes (*Figure 4—figure supplement 2*).

## Porcine adipocytes can be grown in vitro and aggregated into 3D cultured fat

We additionally investigated whether it was possible to grow porcine adipocytes for aggregation into macroscale cultured fat. Primary porcine preadipocytes were observed to readily accumulate lipid (*Figure 5A*). As animal component-free culture media is key to cultivated meat, we also investigated reducing serum in our culture media. Interestingly, sample groups cultured in adipogenic media with 2% FBS contained over 2× fewer cells, yet had over 2× the lipid accumulation, versus groups with 20% FBS (*Figure 5A and B*). Similar effects were seen in preliminary experiments with 0% FBS (*Figure 5—figure supplement 1*). After 30 days of adipogenesis, cultured porcine adipocytes were also successfully combined with alginate to form macroscale cultured fat constructs (*Figure 5C*). When compared to native adipose from food animals (pig, chicken, cow), lipids within cultured porcine adipocytes were multilocular, unlike native adipose from food animals (pig, chicken, cow) (*Figure 5D*).

## In vitro and native pig fat compositions exhibit similarities and can be tuned via lipid supplementation during cell culture

When exploring the effects of Intralipid on primary porcine adipocytes, we observed that lipid supplementation greatly increased their lipid accumulation (*Figure 6—figure supplement 1*). Due to this, we also performed lipidomics to see if primary porcine adipocytes cultured with and without Intralipid also differed in terms of their fatty acid compositions. For the porcine DFAT cells, 500 µg/ml of Intralipid instead of 1000 µg/ml was selected as the test dosage in order to not risk excessive adipocyte cell detachment during differentiation, as a minor increase cell detachment was observed when the cells were treated with 500 µg/ml of Intralipid (data not shown).

Unlike 3T3-L1s, in vitro and native porcine fats exhibited more similar FA profiles, especially within the TAG fractions (*Figure 6A and B*). The only major difference observed between in vitro and native fat samples was a lack of 20:4 (>10% difference) in the phospholipids, with smaller differences

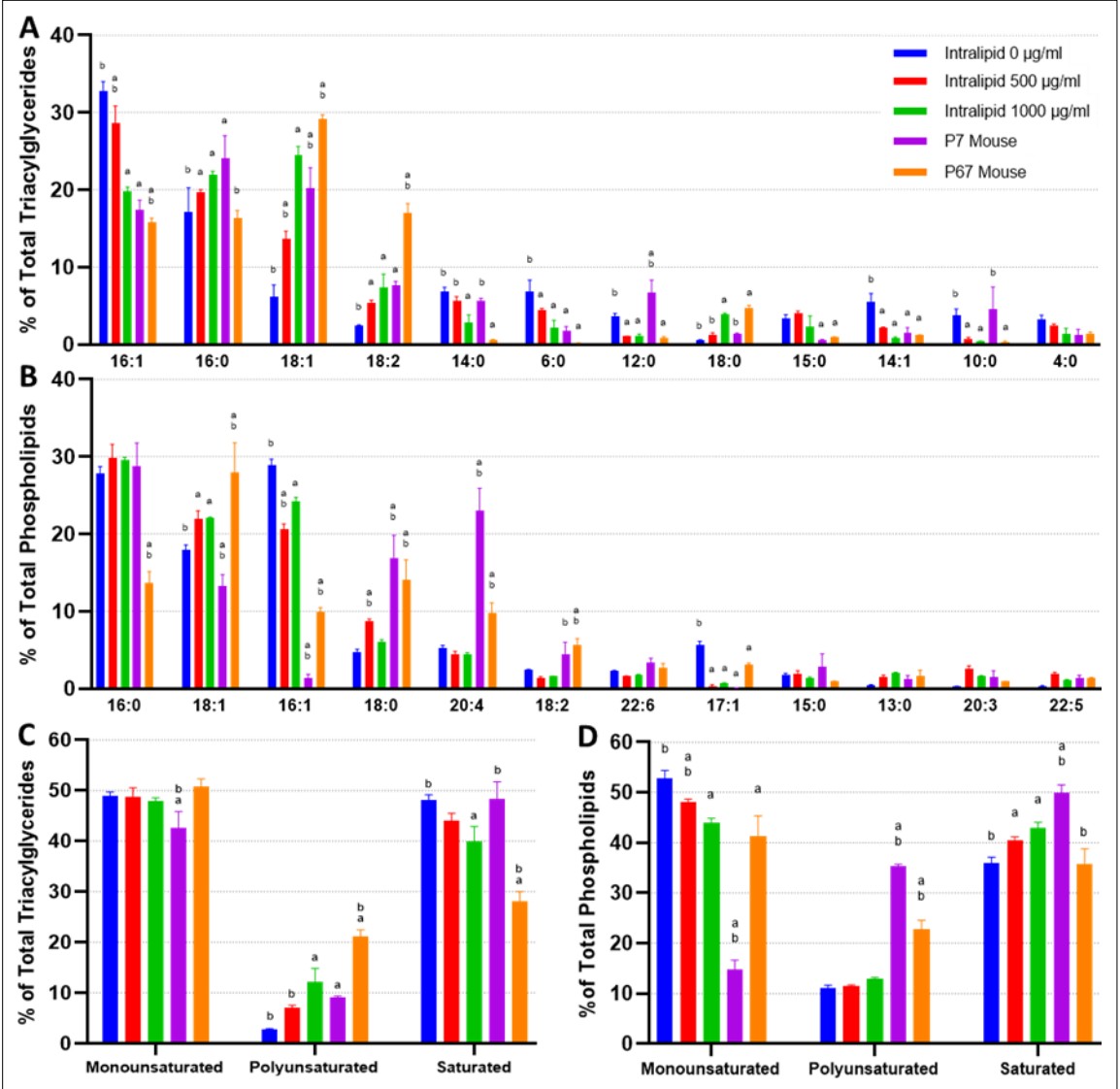

**Figure 4.** Lipidomics analysis of fatty acid compositions in cultured murine adipocytes. (**A, B**) The fatty acid (FA) composition of (**A**) triacylglycerides (TAGs) and (**B**) phospholipids from in vitro (30 days of adipogenesis, 0–1000 µg/ml Intralipid) and native *murine* fats. The stacked graphs on the *left* illustrate the overall FA composition within each sample group (top 6 FAs color coded). The top-down order of FAs shown are the same in the graph and the legend. The column graphs on the *right* show the top 8 most prevalent FAs across all the sample groups with error bars (SD). The top-down order in the legend is the same as the left-right order of the samples in the column charts. Tables containing the proportions of all detected FAs, as well as principal component analysis (PCA) graphs, are available (***Supplementary files 4 and 5***; ***Figure 4—figure supplement 1***). (**C, D**) FA compositions categorized by degree of saturation are shown for (**C**) TAGs and (**D**) phospholipids. The top-down order of monounsaturated (monounsat.), polyunsaturated (polyunsat.), and saturated FAs are the same in the graph and the legend. Column graphs of (**C**) and (**D**) with p≤0.05 comparisons are available in ***Figure 4—figure supplement 3***. (**E**) The 18:3 (TAG fraction) content of 3T3-L1 adipocytes adipogenically cultured for 30 days with various levels of fatty acid (Intralipid) supplementation, versus native mouse fat samples. For all column graphs, 'a' and 'b' represent a difference of p≤0.05 versus 0 µg/ml Intralipid and 1000 µg/ml Intralipid, respectively (analysis of variance [ANOVA] with Tukey's post-hoc tests). 'IN-0', 'IN-500', 'IN-1000' stand for Intralipid 0, 500, 1000 µg/ml Intralipid, respectively. '7D Mouse' and '7M' represent 7-day-old mouse adipose, while '67D Mouse' and '67M' represent 67-day-old mouse adipose. n=3 (technical) for all sample groups.

The online version of this article includes the following figure supplement(s) for figure 4:

**Figure supplement 1.** Principal component analyses of (left) *triacylglyceride* (TAG) and (right) phospholipid fatty acid compositions for in vitro and native *murine* fats (top 30 fatty acids).

**Figure supplement 2.** The proportion of triacylglycerides and phospholipids as a percentage of total intracellular lipids, for in vitro and native *murine* fats.

**Figure supplement 3.** A column graph representation of *murine* fatty acid (FA) saturation in ***Figure 4C and D***.

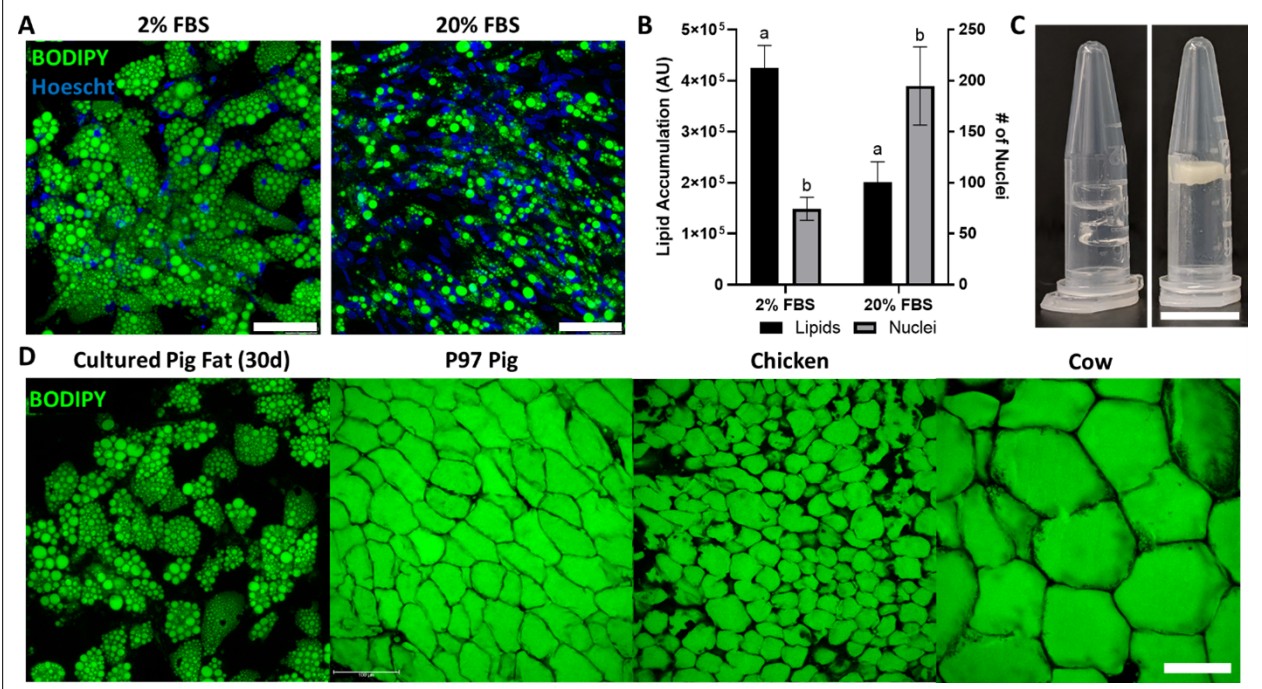

**Figure 5.** Porcine adipocytes grown to produce macroscale cultured fat. (**A**) Pig DFAT cells cultured under adipogenic conditions containing 2% or 20% fetal bovine serum (FBS) for 30 days. Fat cells are stained for lipid using BODIPY (green) and for cell nuclei using Hoescht 33342 (blue). Scale bars 100 μm. (**B**) Lipid and cell number quantification of 30-day pig adipocytes, based on BODIPY and Hoescht 33342 staining, respectively. AU stands for arbitrary units. Results tagged with 'a' represent a difference of p≤0.0001, while 'b' represents p≤0.01. n=4 (technical) for both groups. (**C**, Left) Cell-free 1.6% alginate gels on the left. (**C**, Right) Porcine adipocytes (differentiated in 2% FBS media) mixed with alginate (1.6% final concentration) to form bulk cultured fat. Scale bar represents 1 mm. (**D**) Lipid staining images (BODIPY) of 30-day in vitro porcine adipocytes, juxtaposed with native adipocytes from a 97-day old (97D) pig, as well as a chicken and a cow. Scale bars represent 100 μm.

The online version of this article includes the following figure supplement(s) for figure 5:

**Figure supplement 1.** Pig DFAT cells cultured under adipogenic conditions with 0% fetal bovine serum (FBS) containing media for 30 days.

observed for 18:0, 18:1, 16:0, also within the phospholipid fraction. Additionally, mild deficiencies of 18:2 and 18:3, as well as a slight abundance of 20:1, were observed across the TAG and phospholipid fractions for in vitro samples (versus native porcine fat). Intralipid supplementation appeared to rescue 18:2 and 18:3 levels (while also decreasing 20:1) of in vitro porcine adipocytes, but supplementation also caused FA levels to deviate from native pig fat in other ways (e.g., 15:0, 16:1, 20:3, 20:4, 22:4). These changes can be observed in the PCAs of in vitro and in vivo porcine TAGs and phospholipids, where Intralipid appears to both increase and decrease in vitro adipocyte similarity to native cells depending on the principal component (*Figure 6—figure supplement 2*). When looking at FA saturation however, in vitro and in vivo porcine adipocytes appear to share similar ratios of MUFAs, PUFAs, and SFAs, with Intralipid treatment generally improving the similarity of in vitro and native fats (*Figure 6C and D*).

## Discussion

Our goal was to generate macroscale fat tissue for food applications. To achieve this, we circumvented the limitations of macroscale tissue engineering and 3D cell culture by first culturing murine and porcine adipocytes on 2D surfaces with ample access to the culture media, then aggregating the cells into macroscale 3D tissues after in vitro adipogenesis. This approach is feasible for food applications, as opposed to regenerative medicine needs, because cell viability does not need to be maintained once the final macroscale fat tissue is produced.

We were first able to generate considerable amounts of highly lipid-laden murine adipocytes bearing morphological resemblance to native adipose tissues. Most studies that investigate 3T3-L1 adipogenesis in 2D are of limited culture duration (≤15 days), with lipid accumulation manifesting as small lipid

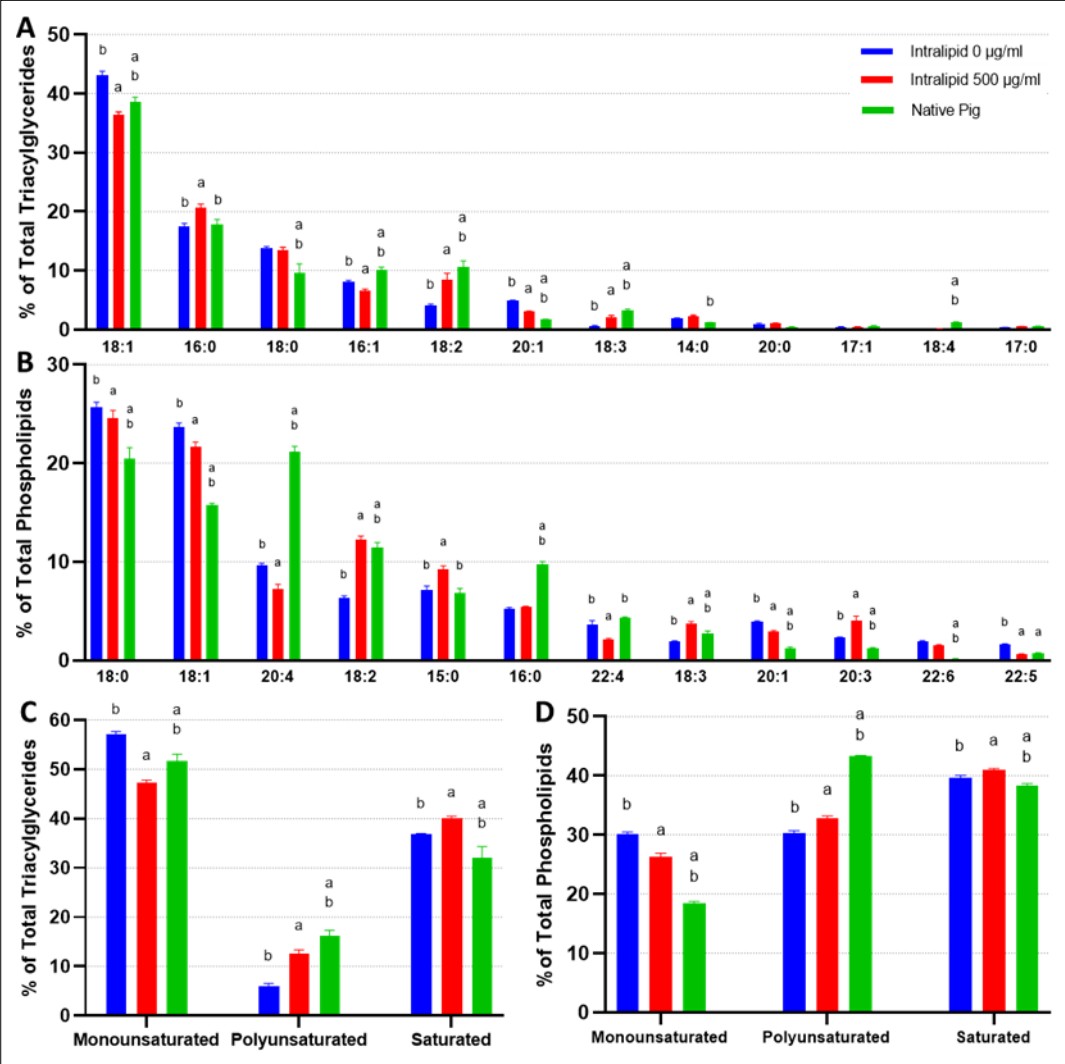

**Figure 6.** Lipidomics analysis of fatty acid compositions in cultured porcine adipocytes. (**A, B**) The fatty acid (FA) composition of (**A**) triacylglycerides (TAGs) and (**B**) phospholipids from in vitro (12 days of adipogenesis, 0–500 μg/ml Intralipid) and native *porcine* fats. The stacked graphs on the *left* illustrate the overall FA composition within each sample group (top 6 FAs color coded). The top-down order of FAs shown are the same in the graph and the legend. The column graphs on the *right* show the top 8 most prevalent FAs across all the sample groups with error bars (SD). 'a' and 'b' represent a difference of p≤0.05 versus 0 μg/ml Intralipid and 500 μg/ml Intralipid, respectively (analysis of variance [ANOVA] with Tukey's post-hoc tests). The top-down order in the legend is the same as the left-right order of the samples in the column charts. Tables containing the proportions of all detected FAs are available in *Supplementary files 6 and 7*. (**C, D**) FA compositions categorized by degree of saturation are shown for (**C**) TAGs and (**D**) phospholipids, displayed similarly as in (**A**) and (**B**). 'IN-0' and 'IN-500' stand for Intralipid 0 and 500 μg/ml Intralipid groups, respectively. 'P' represents native pig adipose. n=3 (technical) for all sample groups. TAGs and phospholipids versus total lipids are available in *Figure 6—figure supplement 3*.

The online version of this article includes the following figure supplement(s) for figure 6:

**Figure supplement 1.** Phase contrast micrographs showing the degree of lipid accumulation in primary porcine adipocytes (DFAT cells) adipogenically differentiated with and without 500 μg/ml Intralipid for 11 days.

**Figure supplement 2.** Principal component analyses of (left) *triacylglyceride* (TAG) and (right) phospholipid fatty acid compositions for in vitro and native *porcine* fats (all fatty acids).

**Figure supplement 3.** The proportion of triacylglycerides and phospholipids as a percentage of total intracellular lipids, for in vitro and native *porcine* fats.

droplets (*Sheng et al., 2014*; *Neal and Clipstone, 2002*; *Sharma et al., 2018*; *Wang et al., 2013*; *Pazienza et al., 2016*; *Bogan et al., 2001*; *Park et al., 2018*). Here, we show that extended 2D culture (30 days) of in vitro adipocytes resulted in the emergence of larger lipid droplets, a key trait of white adipose tissue. Abundant lipids were also observed in porcine adipocytes, especially with 0–2% serum. Enhanced adipogenesis in low serum or serum-free media has been reported by numerous groups; the diminished lipid accumulation found in adipocytes differentiated in high serum conditions (e.g., 20% in this study) may be due to factors in FBS, such as transforming growth factor β, that have been found to be pro-proliferative and anti-adipogenic (*Jurek et al., 2020*; *Mitić et al., 2023*; *Lee et al., 2012*; *Entenmann and Hauner, 1996*; *Sandhu et al., 2017*; *Sprenger et al., 2021*; *Boone et al., 2000*; *Suryawan and Hu, 1993*). Indeed, our results with porcine adipocytes in 20% FBS show increased nuclei and decreased lipid accumulation during differentiation. Despite the abundant lipid accumulation observed in in vitro porcine adipocytes cultured with 0-2% FBS, intracellular lipid accumulation was multilocular. A multilocular phenotype can signify adipocyte browning, indicating the presence of beige or brown fat (*Jung et al., 2019*). However, the lipid morphology observed in the Yorkshire pig DFAT cells of this study likely represents an immature white adipocyte phenotype, as Yorkshire pigs reportedly have no brown adipose tissue (due to the lack of a functional UCP1 gene) (*Zheng et al., 2017*; *Berg et al., 2006*). Optimizations to the in vitro environment and culture regime may help enhance lipid accumulation and promote a unilocular phenotype (*Hsiao et al., 2016*; *Sun et al., 2013*). For example, in this study we found that Intralipid supplementation drastically increased porcine cell adipogenesis. In this study, porcine dedifferentiated fat (DFAT) cells were used as adipocytes as they been reported to proliferate and accumulate lipids for at least 50 passages while maintaining expression of adipocyte-specific genes such as PPARγ (*Nobusue and Kano, 2010*). Additionally, DFAT cells have been shown to accumulate more intracellular lipids than adipose-derived stromal vascular cells, making them a better choice for large-scale cultured fat production (*Peng et al., 2015*).

After in vitro adipocyte generation, mechanical aggregation of the cells during cell scraping immediately produced what looked like fat tissue. Murine adipocytes were then combined with food-relevant binders to maintain aggregation and tissue integrity. Uniaxial compression testing of alginate-based cultured fat tissues revealed similar stress-strain behaviors to native fat tissues, while mTG crosslinked adipocytes exhibited weaker resistance to compression closer to rendered fats. As mTG has been shown to greatly strengthen protein-rich materials, the weaker mTG fat tissues may be due to a lack of extracellular protein available for crosslinking among the aggregated in vitro adipocytes (*Broderick et al., 2005*). Thus, to strengthen mTG fat tissues it may be beneficial to mix in additional proteins such as soy (*Dreher et al., 2020*). It would also be interesting to investigate whether upregulating ECM deposition would improve mTG crosslinking, especially since ECM-promoting culture media components like ascorbic acid have also been shown to improve in vitro adipogenesis (*Jurek et al., 2020*; *Cuaranta-Monroy et al., 2014*; *Liu et al., 2017*; *Kim et al., 2013*; *Thorrez et al., 2018*). Ultimately, the mechanical properties of aggregated cultured fat tissues were tunable based on the binding technique used (alginate versus mTG), as well as via the concentration of binder (0.8 versus 1.6% alginate).

When investigating the TAG compositions of in vitro-grown fats, we found murine 3T3-L1 cells to differ from native fats, notably being deficient in 18:1 and 18:2 while being enriched in 16:1 and shorter chained SFAs. It has been shown that rodents lacking dietary oleic acid (18:1 $\omega$-9) compensated with higher levels of 16:1, as endogenous production of oleic acid is insufficient to achieve the amounts typically found in normal tissue (*Bourre et al., 1997*). Primary porcine adipocytes on the other hand achieved a closer FA profile their native equivalents, though slightly lacking in certain FAs such as 18:2. The scarce presence of 18:2 for both murine and porcine cells is likely due an inability for mammals to synthesize linoleic acid (18:2 $\omega$-6), which may highlight the importance of its supplementation during in vitro culture (*Hansen, 1986*; *Malcicka et al., 2018*). Indeed, Intralipid treatment (44–62% linoleic acid) enriched 18:2 and raised overall PUFA levels for all in vitro adipocytes, with murine cells additionally achieving more similarity to native mouse fats in terms of 16:1 and 18:1 levels (*Abdelghany et al., 2020*). The increase in 18:3 from Intralipid across murine and porcine adipocytes represents a potential enrichment in omega 3 PUFAs, as 18:3 in soybean oil is alpha-linolenic acid. These changes suggest that lipid supplementation may be useful for tuning in vitro adipocyte TAGs to address deficiencies in particular FAs, enhance specific FAs, or more closely match the composition of native fats.

As phospholipids have been reported to contribute significantly to meat flavor, we additionally looked at IVF phospholipid FAs (*Mottram and Edwards, 1983*; *Huang et al., 2010*). In vitro porcine adipocytes were found to contain similar ratios of phospholipid FAs with their in vivo counterparts, although they notably contained fewer 20:4 and 18:2 FAs than the native fat samples tested. As linoleic acid (18:2) is a precursor to arachidonic acid (20:4 $\omega$-6), it was possible that low 20:4 was due to a lack of 18:2. However, Intralipid-treated porcine IVFs did not show increased 20:4 despite having increases in 18:2. 20:4 levels in porcine IVFs were not uncommonly low though, achieving the same value as reported elsewhere for pig fat (1.4% of all FAs) (*Dinh et al., 2021*). High 20:4 in our native pig samples could be related to how native pork fat FAs are very pliable based on the animal's diet (*Wood et al., 2008*). Ultimately, optimal intracellular 20:4 levels are unclear, as arachidonic acid has been linked with both inflammation and enhanced meat flavor (*Calder, 2006*; *Adam et al., 2003*; *Takahashi, 2018*). If desired, raising 20:4 may be possible through supplementation during cell culture, or via genetic interventions (*Takahashi, 2018*; *Rouzer et al., 2006*; *Gol et al., 2018*). For murine IVF phospholipids, Intralipid did not raise 18:2 despite its high linoleic acid content (*Abdelghany et al., 2020*). This could be due to genetic influence over phospholipid FAs and our use of the 3T3-L1 cell line (less representative of native adipocytes). It has been reported that different strains of mice can have varying phospholipid compositions when fed the same diet (*Hulbert et al., 2006*). While useful as a preliminary assessment on the potential flavor or aroma profiles of cultured fat, phospholipid FA profiles are not considered a definitive analysis on fat flavor and aroma. There is thus a need in the field for further research comparing the flavor profiles of cultured fats with native fats and plant-based alternatives, such as via sensory evaluation panels where human participants taste or smell food samples with cultured fat (*Iida et al., 2015*). Gas chromatography/mass spectrometry approaches may also be explored for a quantitative analysis of the various volatile compounds present in the aromas of cultured and in vivo fats (*Watanabe et al., 2008*).

As a result of the comparable fatty acid compositions between in vitro and native fats (especially for primary porcine cells), similar caloric values may be expected for in vitro and native fats on a 'per gram of lipid' basis. However, structured cultured fat tissues in this study may have lowered caloric values due to adipocyte dilution (e.g., the 1:1 mixing of alginate and adipocytes in this study). Here, alternate adipocyte binding protocols and materials may be pursued to create cultured fat tissues with larger proportions of cultured adipocytes. For example, mixing cell-cultured adipocytes with more concentrated alginate would allow a smaller volume to be added for the same final alginate percentage. Other binder materials such as locust bean gum may also be used, which can be dispersed into cultured adipocytes as a powder and then formed into a gel after heating (thus minimally diluting the in vitro-grown adipocytes) (*Dionísio and Grenha, 2012*). With future efforts, characterizations such as calorimetry may be useful for quantifying any differences in caloric value between cultured and in vivo fat tissues. In this realm, thermogravimetric analyses may also be useful for assessing the cooking behavior of cultured fat tissues.

Aggregating in vitro adipocytes to generate cell-cultured animal fat using various binders unlocks a simplified approach of producing bulk cultured fat tissue with lower costs and direct scalability, addressing a key obstacle in cultivated meat production and contributing to the paradigm of food production using cell culture techniques. The adipocyte aggregation approach is cell type and species agnostic, as demonstrated via the 3T3-L1s (adipogenic fibroblasts) and primary porcine DFAT cells (mesenchymal stem cell-like cells) in this study. While laboratory techniques (2D culture in tissue culture flasks) were used to produce the adipocytes for aggregation in this study, the concept of generating adipose tissues for food via the aggregation of individual fat cells should be applicable to cells generated in larger-scale bioreactor systems (*Figure 7*). For example, adherent preadipocytes could be proliferated and differentiated en masse in various fixed bed bioreactors, after which they are harvested (e.g., by spraying fluid onto the cells to detach them) (*Leinonen et al., 2020*; *Pörtner and Platas Barradas, 2007*; *Hanley et al., 2014*; *Leung et al., 2022*; *Yuen et al., 2022*). The large-scale generation of adipocytes might also be achieved in suspension bioreactors using microcarrier, spheroid/aggregate approaches, or through cell adaptation to single cell suspension (*Yuen et al., 2022*; *Frye and Patrick, 2006*; *Lohr et al., 2010*; *Nahmias and Cohen, 2020*; *Bellani et al., 2020*). Once produced, adipocytes can then be formed into structured cultured fat tissues by mixing with binder or crosslinking ingredients. Cultured adipocytes can also be directly added to food products (e.g., soups and sauces) as an unstructured fat biomass. Incorporating cultured fats or fat tissues into

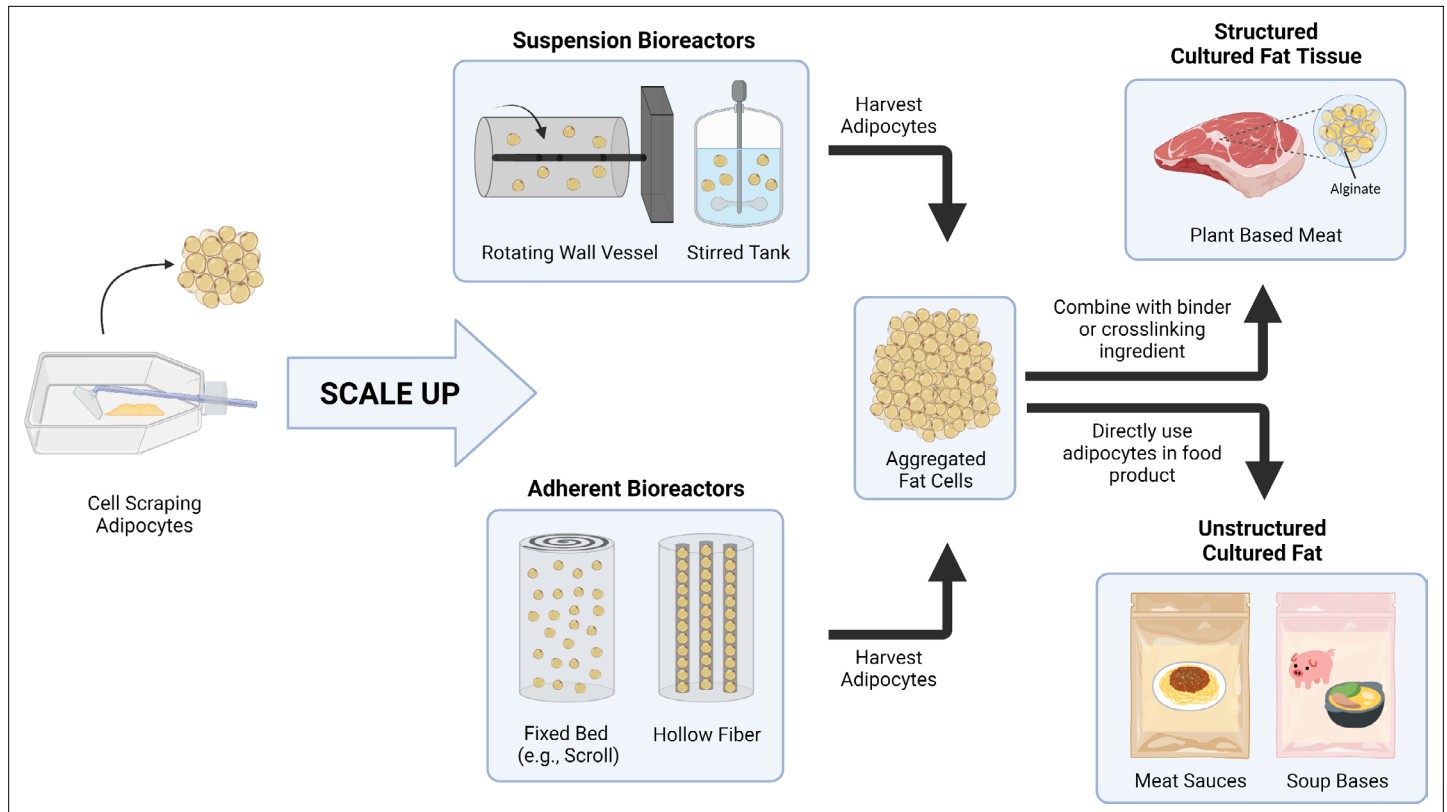

**Figure 7.** A conceptual schematic for scaling up cultured fat production via adipocyte aggregation. As opposed to cell scraping methods with cell culture flasks, large-scale adipose production could utilize different types of bioreactors, including suspension bioreactors (e.g., rotating wall vessel and stirred tank) and adherent bioreactors (e.g., fixed bed, hollow fiber) for cell proliferation and differentiation. The adipocytes are then harvested and aggregated for incorporation into food products. For products requiring structured cultured fat tissues, fat cells are combined with binders or crosslinking ingredients to form a rigid construct. Adipocytes can also be directly included in food products that do not require the cultured fat to be structured.

plant-based meats permits the creation of hybrid products, where sustainable and low-cost plant ingredients are enhanced by the complex species-specific flavors that arise from added animal fat cells. Taken together, these approaches offer a potential path to producing meat (or realistic meat alternatives) without animal slaughter, while potentially being more sustainable than conventional meat production (*Tuomisto et al., 2014*; *Tuomisto and de Mattos, 2011*; *Sun et al., 2015*; *Mattick et al., 2015*; *Odegard and Sinke, 2021*).

# Materials and methods

**Key resources table**

| Reagent type (species) or resource | Designation | Source or reference | Identifiers | Additional information |
|---|---|---|---|---|
| Cell line (*Mus musculus*) | 3T3-L1; murine adipocytes | American Type Culture Collection | CL-173; RRID: CVCL_0123 | |
| Biological sample (*Sus scrofa domesticus*) | DFAT cells; porcine adipocytes | This paper | Subcutaneous fat (pork belly) | Isolation technique outlined in Materials and methods section |
| Biological sample (*Bos taurus*) | Beef fat | This paper | Pectoralis minor (brisket point cut) | Obtained from local butcher shop |
| Biological sample (*Gallus gallus domesticus*) | Chicken fat | This paper | Chicken thigh | Obtained from local butcher shop |

*Continued on next page*

*Continued*

| Reagent type (species) or resource | Designation | Source or reference | Identifiers | Additional information |
|---|---|---|---|---|
| Biological sample (*S. scrofa domesticus*) | Pork fat | This paper | Subcutaneous fat (pork belly) | Obtained from local butcher shop |
| Biological sample (*S. scrofa domesticus*) | Lard (Porcine) | Goya | – | |
| Biological sample (*B. taurus*) | Tallow (Bovine) | Epic Provisions | – | |
| Other | 300 µm cell strainer | pluriSelect | 43-50300-03 | For primary cell isolation (separating cells from debris) |
| Other | Collagenase | Worthington Biochemical | LS004176 | For primary cell isolation (digesting ECM) |
| Chemical compound, drug | DMEM | Thermo Fisher | 10569044 | |
| Chemical compound, drug | Advanced DMEM/F12 | Thermo Fisher | 12634-028 | |
| Chemical compound, drug | GlutaMAX | Thermo Fisher | 35050-061 | |
| Other | Bovine calf serum | Sigma-Aldrich | 12,133C | Culture media component |
| Other | Fetal bovine serum (FBS) | Thermo Fisher | A31606-01; Lot: 2129571 | Culture media component |
| Chemical compound, drug | Antibiotic/antimycotic | Thermo Fisher | 15240062 | |
| Chemical compound, drug | Primocin | InvivoGen | ant-pm-1 | |
| Other | Accumax | Innovative Cell Technologies | AM105 | Cell detachment reagent |
| Other | NucleoCounter | Chemometec | NC-200 | Cell counter |
| Peptide, recombinant protein | Insulin | Sigma-Aldrich | I0516 | |
| Chemical compound, drug | Dexamethasone | Sigma-Aldrich | D4902 | |
| Chemical compound, drug | IBMX | Sigma-Aldrich | I5879 | |
| Chemical compound, drug | Rosiglitazone | TCI America | R0106 | |
| Chemical compound, drug | Intralipid | Sigma-Aldrich | I141 | |
| Chemical compound, drug | Biotin | TCI America | B04631G | |
| Chemical compound, drug | Calcium-D-pantothenate | TCI America | P001225G | |
| Other | Stereolithography 3D Printer | Formlabs | Form 2 | For making custom molds when producing 3D cultured fats |
| Chemical compound, drug | Biocompatible 3D printing resin | Formlabs | Surgical guide resin | |
| Chemical compound, drug | Medical device adhesive | Henkel Adhesives | Loctite 3556 | |
| Chemical compound, drug | Sodium alginate | Modernist Pantry | 1007-50 | |

*Continued on next page*

*Continued*

| Reagent type (species) or resource | Designation | Source or reference | Identifiers | Additional information |
|---|---|---|---|---|
| Chemical compound, drug | Calcium carbonate | Modernist Pantry | 1505-50 | |
| Chemical compound, drug | Glucono delta-lactone | Modernist Pantry | 1159-50 | |
| Peptide, recombinant protein | Microbial transglutaminase (mTG) | Ajinomoto | Activa TI | |
| Other | Goat serum | Thermo Fisher | 16210-064 | For use during immunofluorescent staining |
| Chemical compound, drug | BODIPY 493/503 | Invitrogen | D3922 | |
| Chemical compound, drug | DAPI | Thermo Fisher | 62248 | |
| Chemical compound, drug | Hoescht 33342 | Invitrogen | H3570 | |
| Antibody | Anti-fatty acid synthetase (rabbit, monoclonal) | Invitrogen | MA5-14887; RRID: AB_10980075 | (1:50) |
| Antibody | Anti-peroxisome proliferator-activated receptor gamma (rabbit, polyclonal) | Abcam | ab45036; RRID: AB_1603934 | (1:500) |
| Antibody | Anti-rabbit Alexa Fluor plus 647 (goat, polyclonal) | Invitrogen | A32733; RRID: AB_2633282 | (1:500) |
| Chemical compound, drug | Mounting media | Vector Laboratories | H-1700 | |
| Software, algorithm | Image analysis/quantification | CellProfiler (https://cellprofiler.org) | CellProfiler 4.2.1; RRID: SCR_007358 | |
| Other | Cell scraper (large) | Sarstedt | 83.3952 | For harvest and aggregation of differentiated adipocytes |
| Chemical compound, drug | Methyl tert-butyl ether | Alfa Aesar | 41839AK | |
| Chemical compound, drug | Methanol | Thermo Fisher | BPA4544 | |
| Software, algorithm | Graphing and statistical software | GraphPad Prism (https://www.graphpad.com) | GraphPad Prism 9.3.0; RRID: SCR_002798 | |

## Preadipocyte cell culture

Adipogenic mouse (*Mus musculus*) 3T3-L1 cells were obtained from the American Type Culture Collection (CL-173; ATCC, Manassas, VA, USA) and used to generate cultured fat tissues. 3T3-L1s were received at passage X+13 (with X being an unknown number of passages as stated by ATCC) and expanded in high glucose Dulbecco's modified eagle medium (DMEM) with GlutaMAX, phenol red, and sodium pyruvate (10569044; Thermo Fisher, Waltham, MA, USA) supplemented with 10% bovine calf serum (12,133C; Sigma, Burlington, MA, USA) and 1% antibiotic/antimycotic (Anti/Anti, 15240062; Thermo Fisher) in a 37°C, 5% $CO_2$ incubator. 3T3-L1 cell line identity was confirmed via short tandem repeat profiling, and the cells tested negative for mycoplasma (Case CX4-015723; Labcorp, Burlington, NC, USA) (*Robin et al., 2020*). General cell counting was performed using an automated cell counter (NucleoCounter NC-200; Chemometec, Lillerød, Denmark) and cell detachment was achieved enzymatically using Accumax (AM105; Innovative Cell Technologies Inc, San Diego, CA, USA). For cryopreservation, the same medium was mixed with dimethylsulfoxide (DMSO, D2438; Sigma) at a ratio of 9:1 (medium:DMSO) and cells were frozen overnight at –80°C in an isopropanol-based freezing container (5100-0001; Thermo Fisher). (Long-term storage in liquid nitrogen) 3T3-L1s were used between passages X+15 and X+18.

Dedifferentiated (DFAT) porcine (*Sus domesticus*) cells were generated by isolating mature adipocytes from the subcutaneous adipose tissue of a Yorkshire pig (93 days of age) and was based off several studies on porcine DFAT cells in the literature (*Nobusue and Kano, 2010*; *Peng et al., 2015*). In brief, porcine tissue was procured from the Tufts University Medical Center and transported to the laboratory on ice (~25 min). The tissue was disinfected with a solution of Dulbecco's phosphate buffered saline (14190250; Thermo Fisher) containing 10% Anti/Anti. Five g of tissue was collected from the disinfected tissue and minced into <1 mm$^3$ pieces in the DPBS-Anti/Anti solution. The tissue was then incubated in 25 ml of DMEM containing 0.1% collagenase (LS004176; Worthington Biochemical, Lakewood, NJ, USA) and 10% Anti/Anti for 1.5 hr at 37°C, then 45 min at room temperature (RT). During incubation, the collagenase-adipose solution was inverted every 15 min. Afterwards, the collagenase-adipose solution was filtered through a 300 µm cell strainer (43-50300-03; pluriSelect, San Diego, CA, USA) and centrifuged for 5 min at 500 × *g*. Mature adipocytes were collected from the floating lipid layer in the centrifuged solution and added to 10 ml of DMEM containing 20% FBS (A31606-01, Lot: 2129571; Thermo Fisher) and 100 µg/ml Primocin (ant-pm-1; InvivoGen, San Diego, CA, USA), then centrifuged again (500 × *g*, 5 min). Mature adipocytes were collected and transferred to a T25 tissue culture flask (Nunc EasYFlask, 156367; Thermo Fisher) containing 3.5 ml of DMEM with 20% FBS and 100 µg/ml Primocin and incubated at 39°C with 5% CO$_2$. After 2 days, the floating lipid layer was transferred into a new T25 flask and completely filled (~75 ml) with DMEM+20% FBS+100 µg/ml Primocin. The filled flask was kept at upside down at 39°C, 5% CO$_2$ for 5 days, during which the mature adipocytes attached to the top surface and dedifferentiated. The flask was then drained, flipped around, and used normally with 3.5 ml of culture media. DFAT cells were used up to passage 3, grown using DMEM+20% FBS+100 µg/ml Primocin, and cryopreserved with a 9:1 solution of culture medium and DMSO.

## Adipogenic differentiation

During terminal passages (where cells were differentiated instead of passaged for further cell culture), 3T3-L1s were grown DMEM with 10% FBS and 1% Anti/Anti until 100% confluency. Twenty-four to 48 hr after reaching confluency, cells were switched to an adipogenic induction medium (differentiation medium) consisting of DMEM with 10% FBS, 1% Anti/Anti, 10 µg/ml insulin (I0516; Sigma), 1 µM dexamethasone (D4902; Sigma), 0.5 mM 3-isobutyl-1-methylxanthine (IBMX, I5879; Sigma) and 2 µM rosiglitazone (R0106; TCI America, Portland, OR, USA). After 2 days of adipogenic induction, 3T3-L1s were switched to a lipid accumulation medium for 13–28 days with feeding every 2–3 days. During cell feedings, great care was taken to handle the cells gently to prevent lift-off of lipid-laden adipocytes (*Video 1*). During the aspiration of used culture media, the tissue culture flask is inverted gently to collect media on the top and sidewalls. Dispensing new media (or other liquids) is also performed in the inverted position to avoid direct pipetting on the cells. The flask is then carefully reinverted to its upright position to allow gentle contact with the cells on the bed of the flask. The lipid accumulation medium consisted of DMEM with 10% FBS and 1% Anti/Anti, with 120 µg/ml Intralipid (I141; Sigma) and 10 µg/ml insulin. For experiments on the effects of fatty acid supplementation on cultured adipocytes, Intralipid concentration varied from 0 to 1000 µg/ml. 'Fifteen-day' cultured adipocytes refer to 2 days of adipogenic induction and 13 days of lipid accumulation post-induction, while '30-day' adipocytes refer to 2 days of induction and 28 days of accumulation.

Porcine DFAT cells were differentiated similarly to 3T3-L1s, except the adipogenic induction and lipid accumulation phases lasted for 4 and 26 days, respectively (total 30 days). Different adipogenic media were also used. The induction medium consisted of Advanced DMEM/F12 (12634-028; Thermo Fisher) supplemented with 2 or 20% FBS, 0 or 500 µg/ml Intralipid, 100 µg/ml Primocin, 0.5 µM dexamethasone, 0.5 mM IBMX, 5 µM rosiglitazone, 2 mM (1×) GlutaMAX (35050-061; Thermo Fisher), 20 µM biotin (B04631G; TCI America, Portland, OR, USA), and 10 µM calcium-D-pantothenate (P001225G; TCI America). During lipid accumulation, the same medium was used sans IBMX.

## Harvest of lipid-laden adipocytes and formation of 3D cultured fat tissues

After adipogenic differentiation, adipocytes were rinsed with DPBS after gentle aspiration of the culture media. Flasks were kept on their sides for 5–10 min to more thoroughly drain the DPBS, then cells were harvested using a cell scraper (83.3952; Sarstedt, Nümbrecht, Germany) via a series of

raking motions (as opposed to the windshield wiper approach) into a pre-weighed 50 ml centrifuge tube.

All reconstructed cultured fat tissues were generated in cylindrical 3D printed molds open from the top and bottom (Ø 6 mm, 3 mm height) using a biocompatible resin (Surgical Guide resin; Formlabs, Somerville, MA, USA) and a stereolithography 3D printer (Form2 Printer; Formlabs). A glass coverslip was adhered (Loctite 3556; Henkel Adhesives, Düsseldorf, Germany) to the bottom of the mold to hold the adipocytes during tissue formation and was taken off afterwards to facilitate tissue removal. Two approaches were carried out to form the 3D cultured fat tissues. The first method involved mixing cell-scraped in vitro adipose with a sterile-filtered 15% (w/v) mTG solution (Activa TI; Ajinomoto, Itasca, IL, USA) at a 4:1 ratio (adipose:transglutaminase) and incubating the mixed tissue at 37°C overnight. The second method involved embedding the in vitro adipose in slow gelling alginate of various concentrations, with the approach adapted from the literature (*Growney Kalaf et al., 2016*). Here, 1.6% or 3.2% (w/v) medium viscosity sodium alginate solutions (1007-50; Modernist Pantry, Eliot, ME, USA) were prepared by adding powder to water and allowing the alginate to hydrate for at least 1 hr. After hydration, calcium carbonate powder ($CaCO_3$, 1505-50; Modernist Pantry, Eliot, ME, USA) was added to the alginate solutions to achieve 25 mM and vortexed for 10–60 s to yield evenly dispersed suspensions. To initiate delayed gelation, glucono delta-lactone powder (1159-50; Modernist Pantry, Eliot, ME, USA) was added to the alginate-calcium mixtures to achieve 50 mM and vortexed again for 10–60 s. Finally, the 1.6% and 3.2% alginate solutions were mixed 1:1 (volumetrically) with cell-scraped adipocytes to obtain alginate-cultured fat constructs of 0.8% and 1.6% alginate, respectively. For cultured porcine fat, scraped adipocytes were mixed with 3.2% alginate at a 1:1 (volumetric) ratio. All macroscale cultured fat tissues were stored at 4°C for a maximum of 3 days prior to mechanical testing.

## Native animal adipose samples

Beef fat tissue (*Bos taurus*) from the pectoralis minor (brisket point cut) and chicken fat tissue (*Gallus gallus*) from the thigh were purchased from a local butcher shop. Both samples were displayed at refrigerator temperatures in the shop and transported to the laboratory at RT (~15 min). Two types of pork fat tissues were used in this study. For lipidomics, pork fat from the belly was purchased from a local butcher shop. For mechanical testing and fluorescence imaging, adipose from the belly of a freshly sacrificed 97-day-old Yorkshire pig was obtained at the Tufts University Medical Center and transported to the laboratory on ice (~25 min). Food-grade lard (Goya, Secaucus, NJ, USA) and tallow (Epic Provisions, Austin, TX, USA) were purchased commercially for mechanical testing and shaped with the same 3D printed molds used for forming macroscale IVF tissues. For lipidomics, beef, pork, and chicken samples were flash frozen in liquid nitrogen and stored at –80°C, then thawed for lipid extraction. For mechanical (compressive) testing and fluorescence imaging, beef, pork, and chicken adipose samples were directly used after acquisition. Mouse fat tissue samples were obtained from the perigonadal regions of 7- and 67-day-old mice (CD-1 strain; Charles River, Wilmington, MA, USA) and directly used for lipidomics and fluorescence imaging.

## Mechanical testing

Unconfined uniaxial compressive testing was carried out using a dynamic mechanical analyzer (RSA3; TA Instruments, New Castle, DE, USA). Cultured fat tissue samples (Ø 6 mm, 3 mm height) were taken from storage at 4°C and allowed to warm to RT (25°C as measured by an infrared thermometer) prior to testing. Beef, pork, and chicken fat tissue samples were cut using 6 mm biopsy punches, cut to 3 mm heights, and tested at 25°C. During testing, fat tissue samples were placed between parallel stainless steel plates and compressed. Compressive stress responses and elastic recoveries were measured up to 50% strain at a rate of 1.667% $s^{-1}$. Stress/strain curve data were obtained using the TA Universal Analysis software and represent mean values over n.

## Fluorescence and immunofluorescence staining

Cultured adipocytes in 48-well tissue culture plates were stained to confirm lipid accumulation and to check for markers of adipogenic differentiation. Here, 15- and 30-day 3T3-L1s adipocytes were stained for neutral lipids using DMEM 10% FBS 1% Anti/Anti culture media supplemented with 2 µM BODIPY 493/503 (4,4-difluoro-4-bora-3a,4a-diaza-s-indacene, D3922; Invitrogen, Waltham,

MA, USA). Cells were first rinsed with DPBS, then incubated at 37°C for 15–30 min with 140 µl of the BODIPY-culture media solution in each well. After BODIPY incubation, cells were rinsed twice with DPBS and then fixed in 4% PFA for 10–15 min at RT protected from light. After fixation, cells were rinsed twice in DPBS then simultaneously blocked and permeabilized with B/P buffer (DPBS with 0.1% Tween 20 [P2287; Sigma], 0.1 M glycine [161-0718; Bio-Rad, Hercules, CA, USA], 0.02% sodium azide [S2002; Sigma], and 5% goat serum [16210-064; Thermo Fisher]). Cells were then incubated overnight with primary antibodies at 4°C in B/P buffer containing either 1:50 rabbit anti-fatty acid synthetase (MA5-14887; Invitrogen, Waltham, MA, USA, RRID:AB_10980075) or 1:500 rabbit anti-PPARγ (ab45036, Abcam, Waltham, MA, USA, RRID:AB_1603934). The next day, cells were rinsed in DPBS and incubated with 2 µg/ml DAPI (4',6-diamidino-2-phenylindole, 62248; Thermo Fisher) and 1:500 Alexa Fluor Plus 647 conjugated goat anti-rabbit secondary antibodies (A32733; Invitrogen, Waltham, MA, USA, AB_2633282) for 1 hr at RT. Finally, cells were rinsed in DPBS twice and stored in mounting media (H-1700; Vector Laboratories, Burlingame, CA, USA) at RT. Porcine DFAT adipocytes were stained for lipids via a 15–30 min incubation at 37°C in lipid accumulation media (sans Intralipid) supplemented with 2 µM BODIPY 493/503 and 2 µg/ml Hoescht 33342 (H3570; Invitrogen). DFAT cells were then fixed in 4% PFA at RT for 10–15 min, then rinsed with DPBS and stored in mounting media.

For lipid staining of native adipose samples, chicken, beef, pork, and mouse tissues were first fixed in 4% PFA at 4°C overnight. After fixation, adipose tissues were incubated in optimal cutting temperature compound (OCT, 4583; Sakura Finetek, Torrance, CA, USA) overnight at 4°C. Samples were then placed in molds with additional OCT and flash frozen with liquid nitrogen. Blocks of fat embedded in OCT were then cryosectioned (Leica CM1950; Wetzlar, Germany) and dried on glass slides (Superfrost Plus Gold, 22-035813; Fisher Scientific, Waltham, MA, USA) at RT for 15–30 min. After drying, slides with tissue sections were rehydrated in DPBS for 10 min to dissolve the OCT, then stained with 2 µM BODIPY 493/503 for 30 min at RT. Stained tissues were then cleaned with DPBS and mounted with a coverslip plus mounting media. Staining jars (EasyDip; Simport Scientific, Saint-Mathie-de-Beloeil, QC, Canada) were used whenever possible to minimize sample loss that was sometimes observed when pipetting solutions directly onto the glass slides.

Z-stack images of all samples were taken using a confocal microscope (SP8; Leica, Wetzlar, Germany) using 405, 488, 638 nm lasers as appropriate. Quantifications of lipid accumulation (degree of BODIPY staining) were performed by first segmenting images into stain versus background regions using ilastik, followed by measurements of the stain area in CellProfiler (*Berg et al., 2019*; *Stirling et al., 2021*). BODIPY staining area was divided by the number of nuclei detected to calculate the amount of lipid accumulation per cell. PPARγ % was calculated by counting the number of PPARγ positive nuclei in CellProfiler (after ilastik segmentation), then dividing by total nuclei. Lipid droplet diameters were measured by first segmenting BODIPY images in ilastik, then taking the mean diameter of each BODIPY object (lipid droplet) in CellProfiler.

## Lipid extraction

Lipids were extracted from in vitro adipocytes and native adipose tissues for downstream lipidomics analyses. For this analysis, 10 cm$^2$ worth of in vitro adipocytes were rinsed three times with DPBS with 5 min waits to remove residual culture media, followed by detachment with a cell scraper after the removal of residual DPBS via vacuum aspiration. This cell-scraped cultured fat was used for lipid extractions and lipidomics. Native adipose samples were thawed from –80°C storage (previously flash frozen with liquid nitrogen) and 10 mg (pork, beef, chicken) or 15 mg (mouse) samples were minced with tweezers and scissors prior to lipid extraction. All groups were performed in triplicate.

Lipid extractions were performed using a methyl tert-butyl ether-based (MTBE, 41839AK; Alfa Aesar, Haverhill, MA, USA) approach scaled down for 1.5 ml Eppendorf tubes (*Wong et al., 2019*). Four-hundred µl of methanol (BPA4544; Thermo Fisher) was added to the fat samples in a 1.5 ml centrifuge tube and vortexed for 1 min, followed by an addition of 500 µl MTBE with occasional mixing for 1 hr at RT. Next, 500 µl of water was added and solutions were vortexed for 1 min. Aqueous and organic phases were separated by centrifuging samples at 10,000 × *g* for 5 min. The upper organic phase was then transferred to a new tube while the aqueous phase was re-extracted using 200 µl of MTBE, 1 min of vortexing, and 5 min of centrifugation. The combined organic phases were then dried under nitrogen and stored at –80°C.

## Lipidomics analyses

Lipidomics analyses were performed at the Beth Israel Deaconess Medical Center Mass Spectrometry Facility (John Asara Laboratory) according to prior procedures (*Breitkopf et al., 2017*). In brief, liquid chromatography-mass spectrometry/mass spectrometry (LC-MS/MS) using a Thermo Fisher QExactive Orbitrap Plus mass spectrometer coupled with an Agilent 1200 high pressure liquid chromatography (HPLC) system and an Imtakt Cadenza CD-C18 HPLC column (CD025, 3.0 µm particle size, 2.0 mm inner diameter × 150 mm length) was used to characterize lipid samples. Lipid species identification and peak alignment was achieved using LipidSearch (Thermo Fisher), after which the prevalence of different fatty acids within TAGs and phospholipids were quantified using a custom Python script which is available at: https://github.com/mksaad28/Yuen_et_al; copy archived at *Saad, 2023*.

## Statistical analyses

Statistical analyses were carried out in GraphPad Prism 9.3.0. The specific analyses used are indicated in the text and include: Analysis of variance (ANOVA) with Tukey's post-hoc tests, t-tests with Welsh's correction, PCAs, and chi-square tests. Error bars and ±ranges represent standard deviations unless specified with SEM (standard error of the mean). 0.05 was the cut-off p value for statistical significance. All experiments in this study were carried out with at least triplicate samples (n≥3, technical). The aggregation of 2D grown adipocytes into bulk 3D tissues has been repeated over five times over numerous experiments.

## Acknowledgements

We thank ARPA-E (DE-AR0001233), the NIH (P41EB027062), New Harvest and the United States Department of Defense (DoD) through the National Defense Science and Engineering Graduate (NDSEG) Fellowship Program. We are also grateful to Dr John Asara and the Mass Spectrometry Core Facility at the Beth Israel Deaconess Medical Center for assistance with lipidomics, Sawnaz Shaidani for assistance with BioRender. *Figure 1*, *Figure 7*, and *Figure 3—figure supplement 3* were created with BioRender.com.

## Additional information

### Funding

| Funder | Grant reference number | Author |
| --- | --- | --- |
| New Harvest | Graduate Student Fellowship | John Se Kit Yuen Jr |
| Advanced Research Projects Agency - Energy | DE-AR0001233 | John Se Kit Yuen Jr |
| National Institutes of Health | P41EB027062 | David L Kaplan |
| National Defense Science and Engineering Graduate | Graduate Student Fellowship | John Se Kit Yuen Jr |

The funders had no role in study design, data collection and interpretation, or the decision to submit the work for publication.

### Author contributions

John Se Kit Yuen Jr, Conceptualization, Data curation, Software, Formal analysis, Funding acquisition, Validation, Investigation, Visualization, Methodology, Writing – original draft, Project administration, Writing – review and editing; Michael K Saad, Data curation, Software, Formal analysis, Validation, Visualization, Methodology, Writing – review and editing; Ning Xiang, Investigation, Visualization, Methodology, Writing – review and editing; Brigid M Barrick, Hailey DiCindio, Sabrina W Zhang, Formal analysis, Investigation, Visualization, Writing – review and editing; Chunmei Li, Supervision, Investigation, Methodology, Writing – review and editing; Miriam Rittenberg, Software, Writing – review and editing; Emily T Lew, Investigation, Visualization, Writing – review and editing; Kevin

Lin Zhang, Data curation, Formal analysis, Investigation, Visualization, Writing – review and editing; Glenn Leung, Investigation, Writing – review and editing; Jaymie A Pietropinto, Conceptualization, Resources, Supervision, Funding acquisition, Investigation, Project administration, Writing – review and editing; David L Kaplan, Conceptualization, Resources, Data curation, Formal analysis, Supervision, Funding acquisition, Investigation, Visualization, Project administration, Writing – review and editing

### Author ORCIDs
John Se Kit Yuen Jr ⓘ http://orcid.org/0000-0001-9854-1654
David L Kaplan ⓘ http://orcid.org/0000-0002-9245-7774

### Decision letter and Author response
Decision letter https://doi.org/10.7554/eLife.82120.sa1
Author response https://doi.org/10.7554/eLife.82120.sa2

---

## Additional files

### Supplementary files
• Supplementary file 1. Proportion of nuclei positive for the adipogenic transcription factor peroxisome proliferator-activated receptor gamma (PPARγ) in 3T3-L1 adipocytes grown under adipogenic conditions for 15 days, n=4 technical replicates.

• Supplementary file 2. A frequency distribution table of 15- and 30-day cultured adipocyte lipid droplet diameters compared using a chi-square test (degrees of freedom =8).

• Supplementary file 3. The major fatty acids present in soybean oil (the major component of Intralipid, other than water), according to the package insert supplied with Intralipid 20% (intravenous fat emulsion).

• Supplementary file 4. The full fatty acid compositions (mean % of total of *triacylglycerides* measured, n=3) of in vitro- (30 days of adipogenesis) and in vivo-grown *murine* fats.

• Supplementary file 5. The full fatty acid compositions (mean % of total of *phospholipids* measured, n=3) of in vitro- (30 days of adipogenesis) and in vivo-grown *murine* fats.

• Supplementary file 6. The full fatty acid compositions (mean % of total of *triacylglycerides* measured, n=3) of in vitro- (30 days of adipogenesis) and in vivo-grown *porcine* fats.

• Supplementary file 7. The full fatty acid compositions (mean % of total of *phospholipids* measured, n=3) of in vitro- (30 days of adipogenesis) and in vivo-grown *porcine* fats.

• MDAR checklist

### Data availability
Datasets for our mouse lipidomics data (Figure 3) are included in Supplementary Files 4 and 5. For our porcine lipidomics, datasets are included in Supplementary Files 6 and 7. For lipidomics, this includes tables depicting the top 50 FAs present in our lipidomics analyses for triacylglyceridesTAGs and phospholipids, as well as separate files containing full lists of FAs detected. Our full LC-MS/MS peak aligned and lipid species identified dataset has been uploaded publicly at https://doi.org/10.7910/DVN/DXGQNH.

The following dataset was generated:

| Author(s) | Year | Dataset title | Dataset URL | Database and Identifier |
|---|---|---|---|---|
| Yuen JSK, Saad MK, Xiang N, Barrick MB, DiCindio H, Li C, Zhang SW, Rittenberg M, Lew ET, Leung G, Pietropinto JA, Kaplan DL | 2023 | LipidSearch Dataset for Lipidomics in: Aggregating in vitro-grown adipocytes to produce macroscale cell-cultured fat with tunable lipid compositions for food applications | https://doi.org/10.7910/DVN/DXGQNH | Harvard Dataverse, 10.7910/DVN/DXGQNH |

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
