## [Editor Report]

This paper describes an important new method to cultivate fat tissues in vitro for meat production, by cell aggregation after the adipocytes have fully differentiated in a two-dimensional monolayer. The authors present exceptional evidence that this approach is scalable and that the lipid treatment of the cultured adipocytes modifies their fatty acid composition in the triglyceride as well as the phospholipid portions, which is important for engineering the taste component in cellular agriculture. The work will be of broad interest to bioengineers, tissue engineers, biomaterial scientists, and stem cell biologists.

---

## [Decision Letter]

**Decision letter after peer review:**

Thank you for submitting your article "Aggregating in vitro-grown Adipocytes to Produce Macroscale Cell-Cultured Fat with Tunable Lipid Compositions for Food Applications" for consideration by *eLife*. Your article has been reviewed by 2 peer reviewers, including Milica Radisic as the Reviewing Editor and Reviewer #1, and the evaluation has been overseen by Carlos Isales as the Senior Editor. The following individual involved in the review of your submission has agreed to reveal their identity: Vera M Pieters (Reviewer #2).

Essential revisions:

The authors are encouraged to comment on scalability in terms of reaching the appropriate flat surface area and scraping in industrial applications.

Is it possible to assess the caloric value of these types of products and how far away they are from real products as well as how they change upon thermal processing?

Please improve the statistical analysis.

Provide a rationale for the use of murine cells.

Consider replacing the word fat with fat tissues.

Consider reviewer comments for improving the clarity of the graphical abstract and Figure 1.

The rationale for using dedifferentiated porcine cells as opposed to isolating the adipose-derived stromal mesenchymal cell fraction for adipocyte differentiation is not explained.

*Reviewer #1 (Recommendations for the authors):*

The authors are encouraged to comment on scalability in terms of reaching the appropriate flat surface area and scraping in industrial applications- ie what would be suitable alternative processes?

One of my additional questions would relate to the ability for thermal processing. Is it possible to assess the caloric value of these types of products and how far away they are from real products as well as how they change upon thermal processing?

*Reviewer #2 (Recommendations for the authors):*

Thank you for compiling this interesting work. Below I have outlined a few suggestions and concerns that when addressed could strengthen the manuscript.

I would suggest describing in the figure legends what the 'n' of that specific figure section consists of. For example, an n=4 can describe analyzing 4 images of cells from 4 repeated experiments, or 4 images of the same flask.

The use of 3D 'fats' in the abstract and throughout the text might be confusing to the reader. 'Fats' can also refer to lipid molecules, so using the term '3D fat tissues' might describe your structures better.

The graphical abstract is a great addition to the paper. However, a graphical abstract might be more effective with less text. This graphical abstract almost resembles a conference poster.

The first half of the paper describes the development of macroscale fat tissues and lipid treatment of 3T3-L1 cells. The rationale for using this murine cell line for a paper describing a strategy for developing cultivated fat for food consumption is not well explained.

Line 91: It would be appropriate to describe here what the rationale is for using these binders, as opposed to describing it later in the paper (line 127).

Figure 1E: There are no visible error bars.

Figure 1F and 4D: Scale bars are not visible/not uniform through the different images.

Line 148: The description of 'additionally culture with …' causes confusion. It would be helpful if the authors describe the Intralipid treatment in the context of the timeline described in Figure 1A. The authors could elaborate on the composition of Intralipid and the rationale for using this lipid treatment.

If possible, I would suggest combining Supplemental Figure 4 to Figure 3A.

The text accompanying Figure 3 and Figure 5 is not as easily readable as the rest of the results. The authors could consider maintaining a similar sentence and paragraph structure when describing the results of each of the sub-figures. Such flow might improve readability.

The results of Figure 4A are quite interesting, but not further discussed in this paper. The authors could consider hypothesizing what happens here. Possibly, initially, fewer cells attach in the 2% FBS group after which relatively more lipid from the lipid accumulation media is available to each cell.

Line 212: Lipid accumulation of 33-L1 after Intralipid treatment is not shown.

Line 213: primary PORCINE adipocytes

Line 213: It is unclear what the adipocytes differed from.

Figure 5. It is not clear why the 1000ug/ml Intralipid treatment was not performed here.

Line 247: The authors might be referring to the unlikeliness that they are observing adipose tissue browning in these cultures, however, this might not be clear to an audience not familiar with UCP1 function.

Line 320: The rationale for using dedifferentiated porcine cells as opposed to isolating the adipose-derived stromal mesenchymal cell fraction for adipocyte differentiation is not explained.

---

## [Author Response]

Reviewer #1 (Recommendations for the authors):The authors are encouraged to comment on scalability in terms of reaching the appropriate flat surface area and scraping in industrial applications- ie what would be suitable alternative processes?

Thank you Dr Radisic for your comment – we agree that more discourse on the scalability of the processes outlined in the manuscript are warranted, since large scale production is an emphasis that we make in the text. We have now added a new figure and the following text in the manuscript to cover the topic of cultured fat scale up more holistically:

New Text (Lines 360-374): “While laboratory techniques (2D culture in tissue culture flasks) were used to produce the adipocytes for aggregation in this study, the concept of generating adipose tissues for food via the aggregation of individual fat cells should be applicable to cells generated in larger scale bioreactor systems. For example, adherent preadipocytes could be proliferated and differentiated en masse in various fixed bed bioreactors, after which they are harvested by spraying fluid onto the cells as a means of detaching them^78–81^. The large-scale generation of adipocytes might also be achieved in suspension bioreactors using microcarrier, spheroid/aggregate approaches, or through cell adaptation to single cell suspension^82–86^. Once produced, adipocytes can then be formed into structured cultured fat tissues by mixing with binder or crosslinking ingredients. Cultured adipocytes can also be directly added to food products (e.g., soups and sauces) as an unstructured fat biomass. Incorporating cultured fats or fat tissues into plant-based meats permits the creation of hybrid products, where sustainable and low-cost plant ingredients are enhanced by the complex species-specific flavors that arise from added animal fat cells. Taken together, these approaches offer a potential path to producing meat (or realistic meat alternatives) without animal slaughter, while potentially being more sustainable than conventional meat production^87–91^.”

Figure 7 (Lines 377-383): “A conceptual schematic for scaling up cultured fat production via adipocyte aggregation. As opposed to cell scraping methods with cell culture flasks, large scale adipose production could utilize different types of bioreactors, including suspension bioreactors (e.g., rotating wall vessel and stirred tank) and adherent bioreactors (e.g., fixed bed, hollow fiber) for cell proliferation and differentiation. The adipocytes are then harvested and aggregated for incorporation into food products. For products requiring structured cultured fat tissues, fat cells are combined with binders or crosslinking ingredients to form a rigid construct. Adipocytes can also be directly included in food products that do not require the cultured fat to be structured.”

One of my additional questions would relate to the ability for thermal processing. Is it possible to assess the caloric value of these types of products and how far away they are from real products as well as how they change upon thermal processing?

Thank you for this comment. On a “per gram of lipid” basis, we would expect the same caloric values between in vitro and native fats, due to their similar fatty acid compositions. Conversely, cell-cultured adipocytes mixed with binders such as alginate may have a lowered caloric value due to adipocyte dilution (i.e., the 1:1 mixing of alginate and adipocytes). Alternate protocols or binder materials may permit a closer resemblance to in vivo fats though, should a larger portion of the final cultured fat tissue comprise of the pure cell-cultured adipocytes. An example of this would be mixing cell cultured adipocytes with a more concentrated alginate solution, which would allow for less volume of alginate to be added to the cultured cells to achieve the same final alginate percentage. Another example to maintain a high proportion of adipocytes would be to use another binder such as locust bean gum, which can be dispersed into the cultured fat as a powder, and then formed into a gel after heating.

In the discussion, we have added text to reflect caloric values and analyses such as calorimetry and thermogravimetric analysis, which may be useful as future steps for establishing the equivalence of cultured and native fat tissues (in terms of caloric value and cultured fat tissue cooking behavior).

New Text (Lines 343-354): “As a result of the comparable fatty acid compositions between in vitro and native fats (especially for primary porcine cells), similar caloric values may be expected for in vitro and native fats on a “per gram of lipid” basis. However, structured cultured fat tissues in this study may have lowered caloric values due to adipocyte dilution (e.g., the 1:1 mixing of alginate and adipocytes in this study). Here, alternate adipocyte binding protocols and materials may be pursued to create cultured fat tissues with larger proportions of cultured adipocytes. For example, mixing cell-cultured adipocytes with more concentrated alginate would permit less dilution of in vitro adipocytes a smaller volume to be added for the same final alginate percentage. Other binder materials such as locust bean gum may also be used, which can be dispersed into cultured adipocytes as a powder and then formed into a gel after heating (thus minimally diluting the in vitro-grown adipocytes). With future efforts, characterizations such as calorimetry may be useful for quantifying any differences in caloric value between cultured and in vivo fat tissues. In this realm, thermogravimetric analyses may also be useful for assessing the cooking behavior of cultured fat tissues.”

Reviewer #2 (Recommendations for the authors):Thank you for compiling this interesting work. Below I have outlined a few suggestions and concerns that when addressed could strengthen the manuscript.I would suggest describing in the figure legends what the 'n' of that specific figure section consists of. For example, an n=4 can describe analyzing 4 images of cells from 4 repeated experiments, or 4 images of the same flask.

Thank you for suggesting this. We have updated our figure legends to include these details.

Explanation of technical replicates in the methods section (Lines 571-572): “All experiments in this study were carried out with at least triplicate samples (n ≥ 3, technical).”

Figure 1 New Caption Text (Lines 108-109): “(D) represents n = 4 technical replicates from one experiment, while (E) represents n = 4 technical replicates from a second experiment.”

Figure 2 New Caption Text (Lines 130-132): “n = 4 for all alginate-based cultured fat constructs. n = 3 and 5 for 15-day and 30-day mTG cultured fat constructs respectively. n = 5, 6, 7 for beef, pork, and chicken adipose samples respectively. n = 4 for lard and tallow samples respectively. All ‘n’ values refer to technical replicates.”

Figure 3 New Caption Text (Line 207): “n = 3 (technical) for all sample groups.”

Figure 4 New Caption Text (Line 224): “n = 4 (technical) for both groups.”

Figure 5 New Caption Text (Lines 261-262): “n = 3 (technical) for all sample groups.”

Supplemental Table 1 (Line 881): “n = 4 technical replicates.”

The use of 3D 'fats' in the abstract and throughout the text might be confusing to the reader. 'Fats' can also refer to lipid molecules, so using the term '3D fat tissues' might describe your structures better.

Thank you, we have changed our wording throughout the manuscript and have highlighted where these changes were made.

The graphical abstract is a great addition to the paper. However, a graphical abstract might be more effective with less text. This graphical abstract almost resembles a conference poster.

Thank you for this suggestion. Our previous graphical abstract was deliberately made with higher detail because we thought that typical graphic abstracts were a bit lacking in detail. Nonetheless, we have minimized the text to make things clearer and more concise.

The first half of the paper describes the development of macroscale fat tissues and lipid treatment of 3T3-L1 cells. The rationale for using this murine cell line for a paper describing a strategy for developing cultivated fat for food consumption is not well explained.

Thank you for noting this, we have added additional text explaining this.

New Text (Lines 73-74): “3T3-L1s were chosen for initial experiments as they are readily obtainable from cell banks, and because effective fat differentiation protocols for the cell line already exist.”

Line 91: It would be appropriate to describe here what the rationale is for using these binders, as opposed to describing it later in the paper (line 127).

Thank you for suggesting this, we have moved the explanation for alginate and mTG accordingly.

Figure 1E: There are no visible error bars.

Thank you, we have added error bars to figure 1E.

Figure 1F and 4D: Scale bars are not visible/not uniform through the different images.

Thank you, we have fixed this in the two figures.

Line 148: The description of 'additionally culture with …' causes confusion. It would be helpful if the authors describe the Intralipid treatment in the context of the timeline described in Figure 1A. The authors could elaborate on the composition of Intralipid and the rationale for using this lipid treatment.

Thank you for bringing this up, we have reworded the description and elaborated on the objectives behind screening adipocytes cultured with different amounts of lipid supplementation (the composition of Intralipid is elaborated on further in the text and in Supplemental Table 3).

New Text (Lines 153-159): *“*in vitro adipocytes were cultured with 0 – 1,000 µg/ml of Intralipid (a soybean oil-based lipid emulsion) during the lipid accumulation phase of adipogenic differentiation (d2 – d30). Here, the goal was to investigate any positive or negative effects of lipid supplementation on adipocyte fatty acid (FA) composition relative to in vivo adipocytes*.* Lipid supplementation (e.g., with Intralipid) has been reported to enhance adipogenic differentiation and may thus be a useful strategy in enhancing fat cell lipid accumulation during cultured fat production^44–51^.”

If possible, I would suggest combining Supplemental Figure 4 to Figure 3A.

Thank you, we have added this graph to Figure 3.

The text accompanying Figure 3 and Figure 5 is not as easily readable as the rest of the results. The authors could consider maintaining a similar sentence and paragraph structure when describing the results of each of the sub-figures. Such flow might improve readability.

Thank you for pointing this out. We have reworded text around Figures 3 and 5. We have also revamped the figure with stacked bar graphs to make it more straightforward for readers. We have also adopted a similar format for the captions of Figures 3 and 5.

The results of Figure 4A are quite interesting, but not further discussed in this paper. The authors could consider hypothesizing what happens here. Possibly, initially, fewer cells attach in the 2% FBS group after which relatively more lipid from the lipid accumulation media is available to each cell.

Thank you for suggesting this, we agree that further elaborating on this makes for a better manuscript/paper. We have added additional text regarding the interesting results found with low vs high serum differentiation.

New Text (Lines 276-281): “Enhanced adipogenesis in low serum or serum-free media has been reported by numerous groups; the diminished lipid accumulation found in adipocytes differentiated in high serum conditions (e.g., 20% in this study) may be due to factors in FBS, such as transforming growth factor β (TGF-β), that have been found to be pro-proliferative and anti-adipogenic. Indeed, our results with porcine adipocytes in 20% FBS show increased nuclei and decreased lipid accumulation during differentiation.”

Line 212: Lipid accumulation of 33-L1 after Intralipid treatment is not shown.

Thank you, we have removed the comparison with 3T3-L1s.

Previous Text: “When exploring the effects of Intralipid on primary porcine adipocytes, we observed that lipid supplementation greatly increased their lipid accumulation, in contrast to 3T3-L1s (Supplemental Figure 8).”

New Text (Lines 231-232): “When exploring the effects of Intralipid on primary porcine adipocytes, we observed that lipid supplementation greatly increased their lipid accumulation (Supplemental Figure 9).”

Line 213: primary PORCINE adipocytes. Line 213: It is unclear what the adipocytes differed from.

Thank you, we have modified the sentence here to make things clearer to readers.

Previous Text: “Due to this, we also performed lipidomics to see if primary adipocytes cultured also differed in terms of fat composition.”

New Text (Lines 232-233): “Due to this, we also performed lipidomics to see if primary porcine adipocytes cultured with and without Intralipid also differed in terms of their fatty acid compositions.”

Figure 5. It is not clear why the 1000ug/ml Intralipid treatment was not performed here.

Thank you for this comment. We have added details on our reasoning for the 500 µg/ml Intralipid dosage in the main text.

New Text (Lines 234-237): “For the porcine DFAT cells, 500 µg/ml of Intralipid instead of 1000 µg/ml was selected as the test dosage in order to not risk excessive adipocyte cell detachment during differentiation, as a minor increase cell detachment was observed when the cells were treated with 500 µg/ml of Intralipid (data not shown).”

Line 247: The authors might be referring to the unlikeliness that they are observing adipose tissue browning in these cultures, however, this might not be clear to an audience not familiar with UCP1 function.

Thank you for pointing this out, the text is ultimately better with this additional explanation for readers that may not be familiar with the topic!

Previous Text: “This morphology likely represents an immature white adipocyte phenotype, as Yorkshire pigs lack a functional UCP1 gene^53,54^.”

New Text (Lines 283-286): “A multilocular phenotype can signify adipocyte browning, indicating the presence of beige or brown fat^53^. However, the lipid morphology observed in the Yorkshire pig DFAT cells of this study likely represents an immature white adipocyte phenotype, as Yorkshire pigs reportedly have no brown adipose tissue (due to the lack of a functional UCP1 gene)^54,55^.”

Line 320: The rationale for using dedifferentiated porcine cells as opposed to isolating the adipose-derived stromal mesenchymal cell fraction for adipocyte differentiation is not explained.

Thank you for this comment. We have updated the text to include our reasoning for using dedifferentiated porcine cells as opposed to the stromal mesenchymal cell fraction.

Next Text (Lines 289-293): “In this study, porcine dedifferentiated fat (DFAT) cells were used as adipocytes as they been reported to proliferate and accumulate lipids for at least 50 passages while maintaining expression of adipocyte specific genes such as PPARγ^74^. Additionally, DFAT cells have been shown to accumulate more intracellular lipids than adipose-derived stromal vascular cells, making them a better choice for large scale cultured fat production^75^.”